# Multi-Modal and Multi-Attribute Generation of Single Cells with CFGen

**Alessandro Palma**[1,2]  **Till Richter**[1,2]  **Hanyi Zhang**[1,2]  **Manuel Lubetzki**[1]
**Alexander Tong**[3,4]  **Andrea Dittadi**[1,2,5]  **Fabian J. Theis**[1,2*]
[1]Helmholtz Munich  [2]Technical University of Munich  [3]Université de Montréal
[4]Mila  [5]MPI for Intelligent Systems, Tübingen

## Abstract

Generative modeling of single-cell RNA-seq data is crucial for tasks like trajectory inference, batch effect removal, and simulation of realistic cellular data. However, recent deep generative models simulating synthetic single cells from noise operate on pre-processed continuous gene expression approximations, overlooking the discrete nature of single-cell data, which limits their effectiveness and hinders the incorporation of robust noise models. Additionally, aspects like controllable multi-modal and multi-label generation of cellular data remain underexplored. This work introduces CellFlow for Generation (CFGen), a flow-based conditional generative model that preserves the inherent discreteness of single-cell data. CFGen reliably generates whole-genome, multi-modal, single-cell data, improving the recovery of crucial biological data characteristics while tackling relevant generative tasks such as rare cell type augmentation and batch correction. We also introduce a novel framework for compositional data generation using Flow Matching. By showcasing CFGen on a diverse set of biological datasets and settings, we provide evidence of its value to the fields of computational biology and deep generative models.

## 1 Introduction

Single-cell transcriptomics has revolutionized our ability to study cell heterogeneity, revealing critical biological processes and cellular states (Rozenblatt-Rosen et al., 2017). Advances in single-cell RNA sequencing (scRNA-seq) enable high-throughput gene expression profiling across thousands of cells, providing valuable insights into cellular differentiation (Gulati et al., 2020), disease progression (Zeng & Dai, 2019), and responses to drug perturbations (Ji et al., 2021). Recognizing the complexity of a cell's molecular state, modern studies increasingly integrate additional measurements beyond gene expression, such as DNA accessibility (Grandi et al., 2022) to better characterize gene regulatory mechanisms (Baysoy et al., 2023) or spatially resolved measurements to understand tissue organization (Marx, 2021). Yet, technical bias and high experimental costs still hinder the homogeneous profiling of all possible cell states within the inspected biological process. Generative modeling offers a powerful approach to address these challenges by synthesizing biologically meaningful single-cell data, thereby uncovering underexplored cellular states and improving downstream analyses.

Generative models for single-cell data, in particular Variational Autoencoders (VAEs), have been extensively employed in representation learning (Lopez et al., 2018), perturbation prediction (Lotfollahi et al., 2019; 2023; Hetzel et al., 2022) and trajectory inference (Gayoso et al., 2023; Chen et al., 2022). Recently, more complex approaches leveraging diffusion-based models (Luo et al., 2024) or Generative Adversarial Networks (GAN) (Marouf et al., 2020) have paved the way for the task of synthetic data generation, demonstrating promising performance on realistic single-cell data modeling. Single-cell transcriptome data is inherently discrete, as gene expression is collected as the number of transcribed gene copies found experimentally. Due to the incompatibility of discrete data with continuous models such as Gaussian diffusion (Yang et al., 2023), most approaches generate data pre-processed through normalization and scaling. This limits their flexibility to support downstream tasks centered around raw counts, such as batch correction (Lopez et al., 2018), differential gene expression (Love et al., 2014; Chen et al., 2025; Heumos et al., 2024), and analyses where the

---

*Correspondence to `fabian.theis@helmholtz-munich.de`

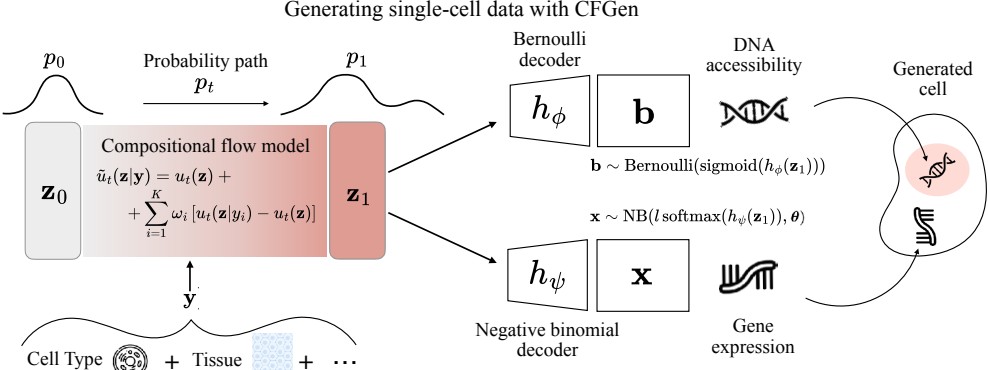

Figure 1: The CFGen generative model. A noise vector $\mathbf{z}_0$ sampled from a Gaussian prior $p_0$ is transformed into a latent cell representation $\mathbf{z}_1$ by a compositional flow, conditioned on multiple biological and technical attributes. Decoders for gene expression and DNA accessibility map $\mathbf{z}_1$ to the parameters of negative binomial and Bernoulli noise models, from which single-cell gene expression and DNA accessibility peaks are sampled.

total number of transcripts in a cell is meaningful (Gulati et al., 2020). Additionally, technical and biological effects in single-cell counts have been formalized under effective discrete noise models (Hafemeister & Satija, 2019), which should be incorporated into generative models for single-cell data to better approximate the underlying data generation process.

In this work, we present CellFlow for Generation (CFGen) (Fig. 1), a conditional flow-based generative model designed to reproduce multi-modal single-cell discrete counts realistically. Our approach combines the expressiveness of recent Flow Matching techniques (Albergo & Vanden-Eijnden, 2023; Liu et al., 2023; Lipman et al., 2023; Dao et al., 2023; Tong et al., 2024) with modeling the statistical properties of single-cell data across multiple modalities, each following a distinct discrete likelihood model. Moreover, we extend the current literature on Flow Matching by introducing the concept of *compositional guidance*, enabling the generation of cells conditioned on single attributes or combinations thereof in a controlled setting.

We evaluate CFGen across multiple biological datasets, demonstrating its advantages in generative performance and downstream applications. Our main contributions are as follows:

- We introduce CFGen, a generative model for discrete multi-modal single-cell data that explicitly accounts for its key statistical properties under a specified noise model.
- We extend the Flow Matching framework to incorporate guidance for compositional generation under multiple attributes.
- We show that our model's full-genome generative performance consistently outperforms existing single-cell generative models qualitatively and quantitatively on multiple biological datasets.
- We showcase the application of CFGen in enhancing downstream tasks, including robust data augmentation for improved classification of rare cell types and batch correction.

## 2 RELATED WORK

The synthetic generation of single-cell datasets is a well-established research direction pioneered by models using standard probabilistic methods to estimate gene-wise parameters in a single modality (Zappia et al., 2017; Li & Li, 2019) or multiple modality setting (Song et al., 2024). With the advent of deep generative models, VAE-based approaches have proven remarkably flexible, offering popular tools for batch correction (Lopez et al., 2018), modality integration (Gayoso et al., 2021), trajectory inference (Gayoso et al., 2023), and perturbation prediction (Lotfollahi et al., 2019; Bereket & Karaletsos, 2023). Despite their relevance, most of the mentioned approaches focus on learning meaningful cellular representations or counterfactual predictions rather than generating synthetic datasets from noise. Such a task has instead been extensively explored by other works leveraging the expressive potential of diffusion models (Luo et al., 2024; Huang et al., 2025), Generative Adversarial Networks (GANs) (Marouf et al., 2020) and Large Language Models (LLMs)

(Levine et al., 2024) to produce realistic cells that approximate the observed data distribution. Our technical contribution builds upon Flow Matching (Albergo & Vanden-Eijnden, 2023; Liu et al., 2023; Lipman et al., 2023), an efficient formulation of continuous normalizing flows for generative modeling. Since its introduction, Flow Matching has been successfully applied to optimal transport (Tong et al., 2024; Eyring et al., 2024; Pooladian et al., 2023), protein generation (Jing et al., 2023; Yim et al., 2023), interpolation on general geometries (Chen & Lipman, 2024; Kapusniak et al., 2025), and guided conditional generation (Zheng et al., 2023). Finally, Flow Matching showed promising performance in tasks involving scRNA-seq, such as learning cellular evolution across time (Tong et al., 2024; Kapusniak et al., 2025) and responses to drugs (Klein et al., 2025).

## 3 BACKGROUND

### 3.1 DEEP GENERATIVE MODELING OF SINGLE-CELL DATA

Single cells are represented as high-dimensional vectors of discrete counts, where each feature corresponds to a gene and its measurement reflects the number of transcripts detected in a cell. Technical bias and biological variation lead to unique characteristics in cells, including *sparsity* and *over-dispersion*. Sparsity arises from genes being inactive in specific cellular states (biological cause) or due to measurement dropouts in scRNA-seq (technical cause). Over-dispersion refers to the presence of greater variance than one would expect from a simple Poisson distribution of the count data (where the gene-wise variance equals the mean). This phenomenon is especially visible in highly expressed genes. Over-dispersed counts are typically modeled using a Negative Binomial (NB) distribution, parameterized by a mean $\mu$ and an inverse dispersion parameter $\theta$. Formally, given a nonnegative count expression matrix $\mathbf{X} \in \mathbb{N}_0^{N \times G}$ with $N$ cells and $G$ genes, entries $x_{ng}$ of the expression matrix are assumed to follow the negative binomial model:

$$x_{ng} \sim \mathrm{NB}(\mu_{ng}, \theta_g) \,, \tag{1}$$

where $\mu_{ng} \in \mathbb{R}_{\geq 0}$ is a cell-gene-specific mean and $\theta_g \in \mathbb{R}_{>0}$ is the gene-specific inverse dispersion. Thus, we assume each cell has an individual mean, while over-dispersion is modeled gene-wise. This parameterization of the negative binomial can be derived from a Poisson-gamma mixture, providing a natural formulation for the scRNA-seq likelihood (see Appendix B.1).

When scRNA-seq is coupled with information on DNA accessibility, transcription measurements are complemented by a binary matrix $\mathbf{B} \in \{0,1\}^{N \times P}$, where $P$ is the number of DNA regions profiled for accessibility measured as the presence (1) or absence (0) of a signal peak. Here, each measurement independently follows the Bernoulli model $b_{np} \sim \mathrm{Bernoulli}(\pi_{np})$, with $\pi_{np}$ indicating a cell-gene-specific success probability.

In most single-cell representation learning settings, a deep latent variable model is trained to map a latent space to the parameter space of the noise model via a decoder maximizing the log-likelihood of the data. Given a latent cell state $\mathbf{z}$, the likelihood parameters of each modality are inferred as

$$\boldsymbol{\mu} = l\boldsymbol{\rho} \,, \quad \boldsymbol{\rho} = \mathrm{softmax}(h_\psi(\mathbf{z})) \,, \quad \boldsymbol{\pi} = \mathrm{sigmoid}(h_\phi(\mathbf{z})) \,, \tag{2}$$

where $h_\psi$ and $h_\phi$ are modality-specific decoders and $l$ is the size factor, defined as the total number of counts of the generated cell. The vector $\boldsymbol{\rho}$ represents gene expression proportions.

### 3.2 CONTINUOUS NORMALIZING FLOWS AND FLOW MATCHING

**Continuous Normalizing Flows (CNF).** Chen et al. (2018) introduced CNFs as a generative model to approximate complex data distributions. Given data in a continuous domain $\mathcal{Z} \subset \mathbb{R}^d$, we define a time-dependent probability path $p : [0,1] \times \mathbb{R}^d \to \mathbb{R}_{\geq 0}$, transforming a tractable prior density $p_0$ into a more complex data density $p_1$, where we indicate the probability path at time $t$ as $p_t : \mathbb{R}^d \to \mathbb{R}_{\geq 0}$ such that $\int p_t(\mathbf{z}) \, d\mathbf{z} = 1$. The probability path is formally *generated* by a time-dependent smooth vector field $u_t : \mathbb{R}^d \to \mathbb{R}^d$, with $t \in [0,1]$, satisfying the continuity equation $\frac{\partial p_t}{\partial t} = -\nabla \cdot (p_t u_t)$. The field $u_t$ is the time-derivative of an invertible *flow* $\phi_t : \mathbb{R}^d \to \mathbb{R}^d$ following the Ordinary Differential Equation (ODE) $\frac{d}{dt}\phi_t(\mathbf{z}_0) = u_t(\phi_t(\mathbf{z}_0))$, where $\phi_0(\mathbf{z}_0) = \mathbf{z}_0$ and $\mathbf{z}_0$ is sampled from $p_0$. The flow $\phi_t$ defines a push-forward transformation $p_t = [\phi_t]_* p_0$, transforming the prior $p_0$ into the data density $p_1$. In other words, learning the vector field $u_t$ that governs the flow allows transporting samples from $p_0$ to $p_1$ by solving the ODE.

**Flow Matching.** Assume the goal is to model a complex data distribution $q$ from a prior $p_0$ by learning a continuous normalizing flow. One can marginalize the probability path $p_t$ as $p_t(\mathbf{z}) = \int p_t(\mathbf{z}|\mathbf{z}_1)q(\mathbf{z}_1)d\mathbf{z}_1$, where $\mathbf{z}_1$ indicates a sample from the data distribution $q$ and $p_t(\cdot|\mathbf{z}_1)$ is a *conditional probability path* transporting noise to $\mathbf{z}_1$ under the boundary conditions $p_0(\mathbf{z}|\mathbf{z}_1) = p_0(\mathbf{z})$ and $p_1(\mathbf{z}|\mathbf{z}_1) \approx \delta(\mathbf{z} - \mathbf{z}_1)$. Here, $\delta$ denotes a Dirac delta measure, which places all probability mass at $\mathbf{z}_1$. Note that, at $t = 1$, the marginal distribution $p_1$ approximates the data distribution $q$. Following the continuity equation, $p_t(\mathbf{z})$ is generated by the *marginal velocity field* $u_t(\mathbf{z})$ that satisfies

$$u_t(\mathbf{z}) = \int u_t(\mathbf{z}|\mathbf{z}_1) \frac{p_t(\mathbf{z}|\mathbf{z}_1)q(\mathbf{z}_1)}{p_t(\mathbf{z})} \, d\mathbf{z}_1, \tag{3}$$

where $u_t(\mathbf{z}|\mathbf{z}_1)$ is called *conditional vector field*. Directly regressing $u_t(\mathbf{z})$ is intractable. However, Lipman et al. (2023) show that minimizing the Flow Matching objective

$$\mathcal{L}_{\text{FM}}(\xi) = \mathbb{E}_{t \sim \mathcal{U}[0,1], q(\mathbf{z}_1), p_t(\mathbf{z}|\mathbf{z}_1)}\left[||v_{t,\xi}(\mathbf{z}) - u_t(\mathbf{z}|\mathbf{z}_1)||^2\right] \tag{4}$$

corresponds to learning to approximate the marginal vector field $u_t$ with the time-conditioned neural network $v_{t,\xi}$ with parameters $\xi$. Defining $p_t(\mathbf{z}|\mathbf{z}_1) = \mathcal{N}(\alpha_t \mathbf{z}_1, \sigma_t^2 \mathbf{I})$ with the functions $\alpha_t, \sigma_t$ controlling the noise schedule, $u_t(\mathbf{z}|\mathbf{z}_1)$ has a closed form, and Eq. (4) is tractable (see Appendix B.2 for more details). We define such a formulation as *Gaussian marginal paths*.

**Classifier-Free Guidance (CFG).** One can *guide* data generation on a condition $y$ by learning the conditional marginal field $u_t(\mathbf{z}|y)$ via a time-conditioned neural network $v_{t,\xi}(\mathbf{z}, y)$. Given a guidance strength hyperparameter $\omega \in \mathbb{R}$, Zheng et al. (2023) show that generating data points following the vector field $\tilde{u}_t(\cdot|y) = (1 - \omega)u_t(\cdot) + \omega u_t(\cdot|y)$ approximates sampling from the distribution $\tilde{q}(\mathbf{z}|y) \propto q(\mathbf{z})^{1-\omega}q(\mathbf{z}|y)^\omega$, where $q(\mathbf{z})$ and $q(\mathbf{z}|y)$ are, respectively, the unconditional and conditional data distributions. The parameter $\omega$ controls the trade-off between diversity and adherence to the condition. This approach enables guidance by interpolating between conditional and unconditional vector fields, both learned jointly during training.

## 4 CFGEN

Our objective is to define a latent Flow-Matching-based generative model for discrete single-cell data, where each cell is measured through gene expression and, potentially, DNA accessibility. Our model, CFGen, is flexible: It can handle single and multiple modalities. Moreover, it supports guiding generation conditioned on single or combinations of attributes without needing to train a different model for each. In what follows, we present the assumptions and generative process formulation in the uni-modal and multi-modal settings. We additionally illustrate our novel approach to compositional guidance.

### 4.1 UNI-MODAL AND SINGLE-ATTRIBUTE GENERATION

Let $\mathbf{X} \in \mathbb{N}_0^{N \times G}$ be a single-cell matrix where an observed single-cell vector is $\mathbf{x} \in \mathbb{N}_0^G$, with $N$ and $G$ being the number of cells and genes. Additionally, let $\mathbf{y} \in \mathbb{N}_0^N$ be a vector of categorical labels associated with each observation. We also define $l = \sum_{g=1}^G x_g$ as the size factor of an individual cell $\mathbf{x}$.

**The generative process.** When the technical bias is negligible, we define the standard CFGen setting as the following generative model:

$$p(\mathbf{x}, \mathbf{z}, l, y) = p(\mathbf{x}|\mathbf{z}, l)p(\mathbf{z}|y, l)p(l)p(y) , \tag{5}$$

where $\mathbf{z}$ is a continuous latent variable modeling the cell state, and we assumed that (1) $\mathbf{x}$ is independent of $y$ conditionally on $\mathbf{z}$ and $l$, and (2) $l$ is independent of $y$. While Eq. (5) defines a standard generative process, the factorization remains flexible based on data properties. Although related to existing VAE-based single-cell generative models, our proposed factorization is novel. We detail the relationship between Eq. (5) and existing generative models in Appendix B.5 and B.6.

**Modeling the distributions in Eq. (5).** Each factor of Eq. (5) is modeled separately: $p(y)$ is a categorical distribution $\text{Cat}(N_y, \boldsymbol{\pi}_y)$ where $N_y$ is the number of categories and $\boldsymbol{\pi}_y$ a vector of $N_y$ class probabilities, and $p(l) = \text{LogNormal}(\mu_l, \sigma_l^2)$. The parameters of $p(y)$ and $p(l)$ can be learned as maximum likelihood estimates over the dataset (see Appendix C.4). Given an attribute class $y$ and size factor $l$ sampled from the respective distributions, $p(\mathbf{z}|y, l)$ is approximated by a conditional continuous normalizing flow $\phi_t(\cdot|y, l)$, with $t \in [0, 1]$, learned via Flow Matching with Gaussian

marginal paths (see Section 3.2, Appendix B.2 and Appendix C.2). Such a flow transports samples $\mathbf{z}_0 \sim \mathcal{N}(\mathbf{0}, \mathbf{I})$ to latent cell representations $\mathbf{z} = \mathbf{z}_1 = \phi_1(\mathbf{z}_0|y, l)$. Let $\mathbf{z}_t = \phi_t(\mathbf{z}_0|y, l)$. The time-derivative of the flow is a parameterized velocity function $v_{t,\xi}(\mathbf{z}_t, y, l)$. Finally, $p(\mathbf{x}|\mathbf{z}, l)$ samples from a negative binomial distribution with mean parameterized by a decoder $h_\psi$ as in Eq. (2) and inverse dispersion modeled by a global parameter $\boldsymbol{\theta}$. In practice, $h_\psi$ and $\boldsymbol{\theta}$ are optimized before training the flow, together with an encoder $f_\eta$ that maps the data to a latent space (more details in Appendix C.1). We outline the reasons for training the encoder and the flow separately in Appendix C.5.

**Sampling in practice.** To generate a cell using CFGen as illustrated in Eq. (5), we first sample a size factor $l$ and a condition $y$ (the latter to specify a class). We then integrate the parameterized vector field $v_{t,\xi}(\mathbf{z}_t, y, l)$ with $t \in [0, 1]$, starting from $\mathbf{z}_0 \sim \mathcal{N}(\mathbf{0}, \mathbf{I})$. We then take the simulated $\mathbf{z}_1 = \phi_1(\mathbf{z}_0|y, l)$ at $t = 1$ as our latent $\mathbf{z}$ in Eq. (5). Finally, we sample $\mathbf{x} \sim \mathrm{NB}(l\,\mathrm{softmax}(h_\psi(\mathbf{z}_1)), \boldsymbol{\theta})$.

**Size factor as a technical effect.** When $l$ is influenced by technical effect under a categorical covariate $c \in \{1, \ldots, C\}$, we reformulate Eq. (5) as $p(\mathbf{x}, \mathbf{z}, l, y) = \frac{1}{C}\sum_c p(\mathbf{x}|\mathbf{z}, l)p(\mathbf{z}|y, l)p(l|c)p(y)p(c)$, where we assume that $\mathbf{z}$ is independent of $c$ given $l$ (i.e., $l$ contains all necessary technical effect information to guide the flow), and $y$ is independent of $c$. The last assumption derives from our choice of $y$ as an attribute encoding biological identity preserved across technical batches.

## 4.2 MULTI-MODAL AND SINGLE-ATTRIBUTE GENERATION

Let $\mathbf{X}$ and $\mathbf{y}$ be defined as in Section 4.1. In the multi-modal setting, we have additional access to a binary matrix $\mathbf{B} \in \{0, 1\}^{N \times P}$ representing DNA region accessibility, with $P$ being the number of measured peaks. Each sample is, therefore, a tuple $(\mathbf{x}, \mathbf{b}, y)$, where $\mathbf{x}$ and $\mathbf{b}$ are realizations of different discrete noise models (negative binomial and Bernoulli). Following Eq. (2), both parameters of the negative binomial and Bernoulli noise models are functions of the same latent variable $\mathbf{z}$, encoding a continuous cell state shared across modalities. We write the first factor in Eq. (5) as

$$p(\mathbf{x}, \mathbf{b}|\mathbf{z}, l) \stackrel{(1)}{=} p(\mathbf{x}|\mathbf{z}, l)p(\mathbf{b}|\mathbf{z}, l) \stackrel{(2)}{=} p(\mathbf{x}|\mathbf{z}, l)p(\mathbf{b}|\mathbf{z}) , \qquad (6)$$

where in (1) we use the fact that the likelihood of $\mathbf{x}$ and $\mathbf{b}$ are optimized disjointedly given $\mathbf{z}$, and in (2) that $\mathbf{b}$ is independent of the size factor $l$ (see Eq. (2)). In simple terms, all the modalities are encoded to the same latent space used to train the conditional flow approximating $p(\mathbf{z}|y, l)$ in Eq. (5) (more details in Appendix C.1). During generation, separate decoders $h_\psi$ and $h_\phi$ map a sampled latent variable $\mathbf{z}$ to the parameter spaces of the negative binomial and Bernoulli distributions, representing expression counts and binary DNA accessibility information, respectively (Fig. 1).

## 4.3 GUIDED COMPOSITIONAL GENERATION WITH MULTIPLE ATTRIBUTES

We extend CFG for Flow Matching (Zheng et al., 2023) to handle multiple attributes, enhancing control over the generative process in targeted data regions. This is especially relevant in scRNA-seq, where datasets are defined by several biological and technical covariates. Here, $\mathbf{Y} \in \mathbb{N}_0^{N \times K}$ represents a matrix of $K$ categorical attributes measured across $N$ cells. Rather than training separate models for each attribute combination, we compose multiple single-attribute flow models.

Let $q(\mathbf{z}|\mathbf{y})$ be the conditional data distribution, with $\mathbf{y} = (y_1, \ldots, y_K)$ being a collection of observed categorical attributes. In analogy with CFG in diffusion models (Ho & Salimans, 2021), we aim to implement a generative model to sample from $\tilde{q}(\mathbf{z}|\mathbf{y}) \propto q(\mathbf{z}) \prod_{i=1}^{K} \left[\frac{q(\mathbf{z}|y_i)}{q(\mathbf{z})}\right]^{\omega_i}$, where $\omega_i$ is the guidance strength for attribute $i$ (see Appendix B.4). Diffusion models generate data by learning to approximate the score of the time-dependent density, $\nabla_{\mathbf{z}} \log p_t(\mathbf{z}|\mathbf{y})$, with a neural network and using it to simulate a reverse diffusion Stochastic Differential Equation (SDE) transporting noise samples to generated data observations (Song et al., 2021). Importantly, the reverse diffusion SDE is associated with a deterministic *probability flow ODE* with the same time-marginal densities (Yang et al., 2023).

CFG in diffusion models can be used to generate data compositionally from different attribute classes. More specifically, Liu et al. (2022) demonstrated that compositional CFG is achievable through parameterizing the drift of the generating reverse SDE with the *compositional score*:

$$\nabla_{\mathbf{z}} \log \tilde{p}_t(\mathbf{z}|\mathbf{y}) = \nabla_{\mathbf{z}} \log p_t(\mathbf{z}) + \sum_{i=1}^{K} \omega_i[\nabla_{\mathbf{z}} \log p_t(\mathbf{z}|y_i) - \nabla_{\mathbf{z}} \log p_t(\mathbf{z})] . \qquad (7)$$

Following the direct relationship between Flow Matching and CFG provided in Ho & Salimans (2021), we build the Flow Matching counterpart to Eq. (7).

Table 1: Quantitative performance comparison of CFGen with conditional and unconditional single-cell generative models. Evaluation is performed based on distribution matching metrics (RBF-kernel MMD and 2-Wasserstein distance). Results are averaged across datasets generated using three different seeds.

| | PBMC3K | | Dentate gyrus | | Tabula Muris | | HLCA | |
|---|---|---|---|---|---|---|---|---|
| | MMD ($\downarrow$) | WD ($\downarrow$) | MMD ($\downarrow$) | WD ($\downarrow$) | MMD ($\downarrow$) | WD ($\downarrow$) | MMD ($\downarrow$) | WD ($\downarrow$) |
| Conditional | | | | | | | | |
| c-CFGen | **0.85** ± 0.05 | **16.94** ± 0.44 | **1.12** ± 0.04 | **21.55** ± 0.17 | **0.19** ± 0.02 | **7.39** ± 0.20 | **0.54** ± 0.02 | **10.72** ± 0.08 |
| scDiffusion | 1.27 ± 0.20 | 22.41 ± 1.21 | 1.22 ± 0.05 | 22.56 ± 0.10 | 0.24 ± 0.04 | 7.89 ± 0.45 | 0.96 ± 0.04 | 15.82 ± 0.45 |
| scVI | 0.94 ± 0.05 | 17.66 ± 0.29 | 1.15 ± 0.04 | 22.61 ± 0.23 | 0.26 ± 0.02 | 9.76 ± 0.53 | 0.58 ± 0.02 | 11.78 ± 0.19 |
| Unconditional | | | | | | | | |
| u-CFGen | 0.44 ± 0.01 | 16.81 ± 0.06 | 0.42 ± 0.01 | **21.20** ± 0.02 | **0.08** ± 0.00 | **8.54** ± 0.06 | **0.15** ± 0.01 | **10.63** ± 0.01 |
| scGAN | **0.36** ± 0.01 | **15.54** ± 0.06 | 0.42 ± 0.01 | 22.52 ± 0.03 | 0.25 ± 0.00 | 12.85 ± 0.04 | 0.18 ± 0.01 | 10.81 ± 0.01 |

Provided that we have access to Flow Matching models for the unconditional marginal vector field $u_t(\mathbf{z})$ and the single-attribute conditional fields $\{u_t(\mathbf{z}|y_i)\}_{i=1}^K$, the following holds:

**Proposition 1** *If the attributes $y_1, ..., y_K$ are conditionally independent given $\mathbf{z}$, the vector field*

$$\tilde{u}_t(\mathbf{z}|\mathbf{y}) = u_t(\mathbf{z}) + \sum\nolimits_{i=1}^K \omega_i[u_t(\mathbf{z}|y_i) - u_t(\mathbf{z})] \tag{8}$$

*coincides with the velocity of the probability-flow ODE associated with the generative SDE of a diffusion model with a compositional score as in Eq. (7).*

We provide a proof for Proposition 1 in Appendix B.4. In other words, the reversed diffusion SDE from compositional CFG admits a deterministic probability flow ODE with velocity as in Eq. (8). Consequently, CFG sampling from compositions of attributes is obtained by integrating the parameterized field $\tilde{v}_{t,\xi}(\mathbf{z}, \mathbf{y}) = v_{t,\xi}(\mathbf{z}) + \sum_{i=1}^K \omega_i[v_{t,\xi}(\mathbf{z}, y_i) - v_{t,\xi}(\mathbf{z})]$ starting from samples from a Gaussian prior $p_0$. Note that both conditional and unconditional fields are parameterized by the same model, which is learned by providing single-attribute conditioning with a certain probability during training (Algorithms 1 and 2).

## 5 EXPERIMENTS

In this section, we compare CFGen with existing models in uni-modal (Section 5.1) and multi-modal (Section 5.2) generation across five datasets. We evaluate quantitatively by measuring distributional proximity between real and generated cells, and qualitatively by assessing how well models capture real data properties. In Section 5.3, we show the effectiveness of multi-attribute generation in guiding synthetic samples towards specific biological labels and donors. Lastly, in Sections 5.4 and 5.5, we demonstrate that CFGen enhances rare cell type classification through targeted data augmentation and performs batch correction on par with widely used VAE-based models.

### 5.1 COMPARISON WITH EXISTING METHODS ON UNI-MODAL SCRNA-SEQ GENERATION

We evaluate the performance of CFGen conditionally and unconditionally against three baselines.

**Baselines.** We choose scVI (Gayoso et al., 2021) and scDiffusion (Luo et al., 2024) as conditional models and scGAN (Marouf et al., 2020) as unconditional baseline. The scVI model is based on a VAE architecture with a negative binomial decoder and performs generation by decoding low-dimensional Gaussian noise into parameters of the likelihood model. Conversely, scDiffusion and scGAN operate on a continuous-space domain, performing generation using standard latent diffusion (Rombach et al., 2022) and GAN (Goodfellow et al., 2014) models. Thus, we train them using normalized counts (more in Appendix D).

**Datasets.** We assess model performance on four datasets of varying size: **(i)** PBMC3K[1] (2,638 cells from a healthy donor, clustering into 8 cell types), **(ii)** Dentate gyrus (La Manno et al., 2018) (18,213 cells from a developing mouse hippocampus), **(iii)** Tabula Muris (Tabula Muris Consortium et al., 2018) (245,389 cells from *Mus musculus* across multiple tissues), and **(iv)** Human Lung Cell Atlas (HLCA) (Sikkema et al., 2023) (584,944 cells from 486 individuals across 49 datasets). Conditioning

---

[1] https://satijalab.org/seurat/articles/pbmc3k_tutorial.html

is performed on *cell type* for all datasets except Tabula Muris, where we use the *tissue* label. Dataset descriptions and pre-processing details are in Appendix E and Table 4.

**Quantitative evaluation.** As evaluation metrics, we use distribution distances (RBF-kernel Maximum Mean Discrepancy (MMD) (Borgwardt et al., 2006) and 2-Wasserstein distance) computed between the Principal Component (PC) projections of generated and real test data in 30 dimensions. The generated data is embedded using the PC loadings of the real data for comparability. For conditional models, we evaluate the metrics per cell type and average the results. All evaluations are performed on a held-out set of cells, considering the whole genome, with a filtering step for low expression genes (see Appendix E).

**Quantitative results.** In Table 1, we evaluate the generative performance of CFGen conditionally (c-CFGen) and unconditionally (u-CFGen) against the three baselines on the scRNA-seq generation task. CFGen consistently reaches the highest performance on conditional generation across biological categories and overcomes scGAN on three out of four datasets on the unconditional generation task.

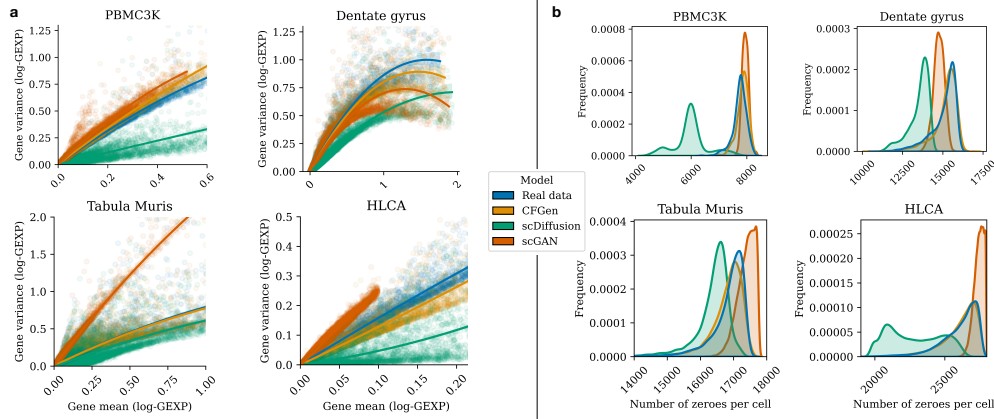

Figure 2: **(a)** Comparison between the gene-wise empirical mean-variance trend in real data and samples from generative models. **(b)** Frequency of the number of zeroes per cell in real and generated data.

**Qualitative evaluation.** Evaluating realistic data generation in biology requires more than distribution-matching metrics. We compare CFGen with diffusion and GAN-based single-cell generators on the task of modeling the probabilistic properties of the single-cell data. Specifically, we consider how well different methods recover the following aspects from real gene expression counts: (1) Sparsity: caused by technical biases in gene transcript detection or gene inactivity in specific contexts. (2) Over-dispersion: a nonlinear mean-variance relationship, modeled through the inverse dispersion parameter of a negative binomial distribution. (3) Discreteness.

**Qualitative results.** In Fig. 2, we provide qualitative evidence that CFGen is more effective in recovering properties (1) and (2) compared to scDiffusion and scGAN, which assume a continuous data space. Property (3) naturally follows when modeling discrete counts with CFGen. Specifically, Fig. 2a shows that explicitly modeling counts with gene-specific inverse dispersion leads to better alignment of the generated gene-wise mean-variance relationship with real data. Additionally, Fig. 2b demonstrates the recovery of the actual distribution of zero counts per cell. In contrast, scDiffusion often shifts towards actively expressed genes, while scGAN tends to either under or overestimate data sparsity. Furthermore, CFGen is the only conditional model capable of generating plausible synthetic cells in terms of overlap with the real data distribution for large datasets such as the HLCA and Tabula Muris (see Appendix H.3 and Fig. A6).

## 5.2 MULTI-MODAL GENERATION

We evaluate the qualitative and quantitative performance of CFGen at generating multi-modal data comprising gene expression and binary DNA-region accessibility.

**Baselines.** We compare CFGen with a VAE-based multi-modal generative model (MultiVI) (Ashuach et al., 2023). For completeness, we add as baselines two single-modality generative models: PeakVI (Ashuach et al., 2022) (DNA accessibility) and scVI (Lopez et al., 2018) (gene expression). Finally, we include scDiffusion and uni-modal CFGen (CFGen RNA) as baselines for scRNA-seq generation.

**Datasets.** We use the multiome PBMC10K dataset, made available by 10X Genomics [2]. Here, each cell is measured both in gene expression (RNA) and DNA accessibility (ATAC). The dataset consists of 10K cells across 25,604 genes and 40,086 peaks and was annotated with 15 cell types, with their respective marker peaks (enriched in accessible or inaccessible points) and genes.

**Evaluation.** We use the RBF-kernel MMD and 2-Wasserstein distances in the same setting described in Section 5.1. Before comparison, we normalize both real and generated binary measurements of DNA accessibility via TF-IDF (Aizawa, 2003) (in analogy to text mining). The metrics are computed in a 30-dimensional PC projection of the generated cells, using the PC loadings of the real data. RNA counts are treated as in Section 5.1. In Appendix H.4 we compare CFGen and MultiVI more biologically. Specifically, we assess how well they approx-

Table 2: Comparison between CFGen, scDiffusion and VAE-based models on generating multiple single-cell modalities. We report distribution distance performance (RBF-kernel MMD and 2-Wasserstein distance) between real and generated cells across three seeds. Underlined values indicate the second-best performance.

|  | RNA | | ATAC | |
|---|---|---|---|---|
|  | MMD ($\downarrow$) | WD ($\downarrow$) | MMD ($\downarrow$) | WD ($\downarrow$) |
| CFGen multi. | $\underline{0.89}_{\pm 0.02}$ | $\mathbf{13.90}_{\pm 0.07}$ | $\mathbf{0.92}_{\pm 0.02}$ | $\mathbf{18.86}_{\pm 0.37}$ |
| CFGen RNA | $\mathbf{0.86}_{\pm 0.02}$ | $\underline{14.30}_{\pm 0.08}$ | - | - |
| scDiff. | $1.02_{\pm 0.02}$ | $14.82_{\pm 0.11}$ | - | - |
| MultiVI | $\mathbf{0.86}_{\pm 0.03}$ | $15.92_{\pm 0.25}$ | $\underline{0.96}_{\pm 0.03}$ | $21.09_{\pm 0.34}$ |
| PeakVI | - | - | $1.49_{\pm 0.02}$ | $\underline{20.84}_{\pm 0.45}$ |
| scVI | $0.95_{\pm 0.02}$ | $14.38_{\pm 0.11}$ | - | - |

imate per-cell-type marker peaks and gene expression (see Appendix F.4). For each cell type, we compute the accessibility fraction and mean expression of literature-derived marker peaks and genes in both real and generated cells and report their correlation per cell type in Fig. A7b.

**Results.** CFGen outperforms both MultiVI and PeakVI in modeling accessibility data based on distribution matching metrics (see Table 2). When considering the RNA modality, our model surpasses scVI and scDiffusion in all metrics and MultiVI in terms of 2-Wasserstein distance. Qualitatively, Fig. A7a shows substantial overlap between real and generated modalities. Finally, Fig. A7b demonstrates that CFGen better approximates average marker peak accessibility and gene expression, outperforming MultiVI across all cell type categories.

## 5.3 MULTI-ATTRIBUTE GENERATION AND GUIDANCE

We assess our approach to compositional guidance, as outlined in Section 4.3.

**Datasets.** We showcase guidance on datasets with extensive technical variation, as one could combine different levels of biological and technical annotations to either augment rare cell type and batch combinations or control for the amount of technical effect added in simulation settings. Specifically, we consider **(i)** The NeurIPS 2021 dataset (Luecken et al., 2021a) - 90,261 bone marrow cells from 12 healthy human donors. We use donor as a batch attribute and cell type as a biological covariate. We also consider **(ii)** the Tabular Muris dataset described in Section 5.1, using tissue and Mouse ID as covariates.

**Evaluation.** The power of our guidance model is to generate data conditionally on an arbitrary subset of attributes—including unconditional generation—using a *single trained model*. For each pair of covariates $(y_i, y_j)$, we evaluate generation on 500 generated cells varying the parameter $\omega_j$, keeping $\omega_i$ fixed. The expected result is conditional generation on $\omega_i$ when $\omega_j = 0$ and generation from the intersection between the two attributes as $\omega_j$ increases. We additionally test the unconditional model, given by $\omega_i = \omega_j = 0$, expected to recover the whole single-cell dataset. In the unconditional case, we generate as many cells as there are in the dataset to better evaluate the coverage.

**Results.** Visual guidance results are shown in Fig. 3, with examples of double-attribute guidance between CD14+ monocytes and donor 1 for the NeurIPS 2021 dataset and tongue and mouse 18-M-52 for Tabula Muris. Unconditional generation recreates the original data (left-hand side) for both datasets. Setting guidance weights to zero for mouse ID and donor attributes leads to single-attribute conditional generation. Increasing the guidance weight steers the generation to the intersection of the two attributes. Quantitative results on attribute intersection generation quality are in Appendix H.7.

## 5.4 APPLICATION: DATA AUGMENTATION TO IMPROVE CLASSIFICATION OF RARE CELL TYPES

We explore using CFGen to improve cell-type classifier generalization by augmenting rare cell types in datasets. Previous work has shown data augmentation enhances cell type classification (Richter et al., 2024). As a classifier, we use scGPT (Cui et al., 2024), a transformer pre-trained on 33 million cells.

---

[2] https://www.10xgenomics.com/support/single-cell-multiome-atac-plus-gene-expression

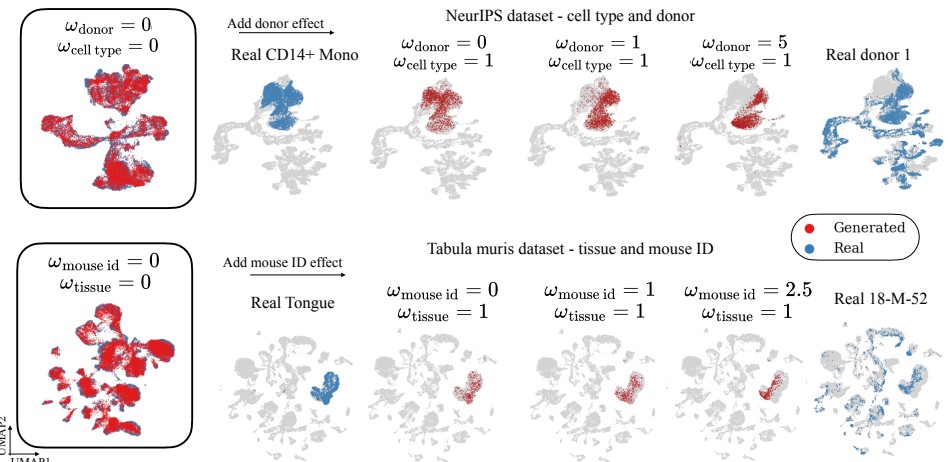

Figure 3: Qualitative evaluation of guidance performance on attribute pairs in the NeurIPS 2021 and Tabula Muris datasets. Left: unconditional performance with guidance weights at 0. Moving right: simulate 500 cells, progressively increasing the guidance strength of one attribute while keeping the counterpart unchanged.

**Datasets.** We leverage two large datasets: **(i)** PBMC COVID (Yoshida et al., 2022) - 422,220 blood cells from 93 patients ranging across paediatric and adult. **(ii)** The HLCA dataset described in Section 5.1. Both datasets are processed by selecting 2000 highly variable features and holding out cells from 20% of the donors.

**Evaluation.** We train CFGen on the PBMC COVID and HLCA training sets and successively augment both to 800,000 samples by upsampling rare cell types. For each cell type, we compute $\frac{1}{N_{ct}}$, where $N_{ct}$ is the total number of cells from a cell type ct. We then generate observations to fill the gap between the dataset size and 800,000 cells, sampling cells proportional to the inverse of their cell type frequency. This process yields significantly more observations for rare cell types. However, we still do not reach uniformity, as class imbalance may be biologically meaningful. Following the original publication, we train kNN cell-type classifiers on scGPT's embeddings from the original and augmented training sets, evaluating the recall

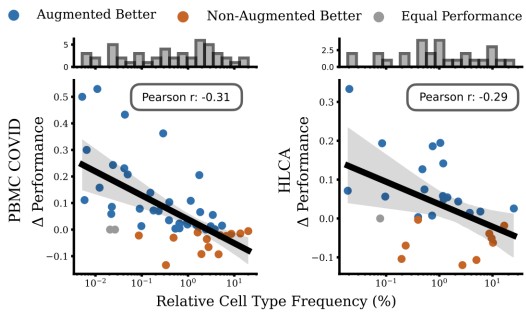

Figure 4: Cell-type classification recall difference before and after augmentation as a function of cell type frequency. The classifier is a 10-nearest neighbor (kNN) model trained on the scGPT's representation space.

performance on held-out donors. For each cell type, we assess if performance increases upon augmentation as a function of its frequency in the dataset.

**Results.** Our results are displayed in Fig. 4 for the two datasets. Remarkably, most cell types in the held-out dataset are better classified after augmentation, suggesting that CFGen not only generates reliable cell samples but can be a valuable supplement to relevant downstream tasks. Moreover, the performance difference between before and after augmentation is inversely proportional to the frequency of the cell type in the dataset. Therefore, the improvement in generalization is more accentuated for rare cell types. Finally, Fig. A10 in the Appendix shows that augmentation via CFGen outperforms the competing methods at improving the generalization performance on rare cell types in unseen donors. We provide raw cell type recall metrics in Table 7 and Table 8.

## 5.5 APPLICATION: BATCH CORRECTION

We apply multi-attribute CFGen to batch correction (see Fig. 5), a common use case for generative models in scRNA-seq (Tran et al., 2020; Luecken et al., 2021b). Given a dataset with batch labels, we choose a reference batch $y_{batch}^{ref}$. For a latent cell representation $\mathbf{z}_j$ with attributes $y_{batch}^{(j)}$ and $y_{cell\,type}^{(j)}$, we invert the generative flow to remove the attribute structure. Next, we simulate the forward flow from the obtained representations while fixing the cell type and assigning $y_{batch}^{ref}$ to all observations. Guidance weights regulate the preservation of biological versus batch labels.

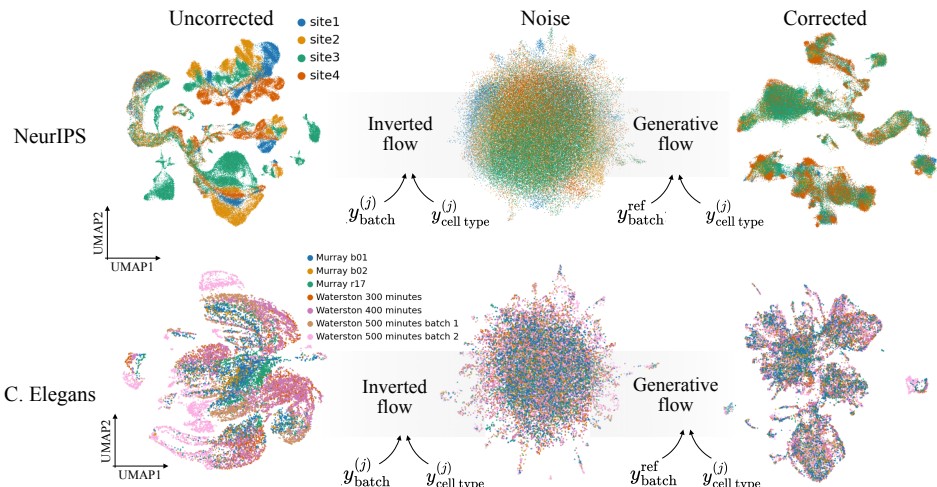

Figure 5: To perform batch correction, the scRNA-seq latent distribution is mapped to the prior distribution by inverting the flow model. The resulting points are then transported back to the data domain based on a common reference batch label and the original cell type label to preserve the biological structure. Cells are colored by batch.

**Datasets.** We evaluate CFGen as a batch correction method on two datasets: **(i)** The NeurIPS dataset described in Section 5.3, using *cell type* as a biological variable to preserve and *acquisition site* as batch variable. **(ii)** The C. Elegans molecular atlas (Packer et al., 2019), which profiles 89,701 cells across 7 sources (batches). Similarly to (i), we use cell type as a biological annotation.

Table 3: Average batch correction and biological conservation metrics from the scIB package comparing CFGen with VAE-based batch correction models in a 50-dimensional representation space. PC projections of the data are used to evaluate the uncorrected data.

| | NeurIPS | | C. Elegans | |
|---|---|---|---|---|
| | Batch (↑) | Bio (↑) | Batch (↑) | Bio (↑) |
| CFGen | **0.63** | 0.61 | **0.68** | **0.63** |
| scPoli | 0.55 | 0.64 | 0.61 | 0.56 |
| scANVI | 0.48 | **0.68** | 0.61 | 0.59 |
| scVI | 0.45 | 0.63 | 0.58 | 0.55 |
| Uncorrected | 0.33 | 0.62 | 0.40 | 0.53 |

**Evaluation.** We compare our model with established VAE-based integration methods: scANVI (Xu et al., 2021), scVI, and scPoli (De Donno et al., 2023). Using scIB metrics (Luecken et al., 2021b), we assess batch correction and biological conservation based on neighborhood composition in the embedding space (see Appendix F.7). All methods are evaluated on a 50-dimensional latent space, with scores from the PC representation of uncorrected data included for comparison. We find that setting $\omega_{\text{batch}} = 1$ and $\omega_{\text{cell type}} = 2$ for C.Elegans and $\omega_{\text{batch}} = 2$ and $\omega_{\text{cell type}} = 1$ for NeurIPS preserves cell type variation while correcting for technical variation (see Appendix H.8 for more details on the selection).

**Results.** We present our technical effect correction approach alongside qualitative results in Fig. 5, illustrating batch mixing performance across both datasets. In Table 3, CFGen is benchmarked against baseline methods. Our model achieves superior batch correction, surpassing the second-best approach by 8% on the NeurIPS dataset and 7% on C. Elegans. On the latter dataset, CFGen additionally outperforms baseline models in preserving biological structure after correction.

## 6 CONCLUSION

We presented CFGen, a conditional latent flow-based generative model for single-cell discrete data that combines state-of-the-art generative approaches with rigorous probabilistic considerations. CFGen incorporates established noise models to sample realistic gene expression and DNA accessibility states, with promising applications in data augmentation and batch correction tasks. Furthermore, our model demonstrates improved performance over existing generative frameworks, reproducing data more faithfully across modalities. Our core machine learning contribution extends classifier-free guidance in Flow Matching with compositional generation of multiple attributes. Overall, CFGen represents a significant advancement in the simulation and augmentation of single-cell data, offering the research community powerful tools to support biological analysis.

**Limitations.** Our framework relies on multiple assumptions, including independence in the data, which may not hold in all biological contexts. Thus, exploring data characteristics is essential before using CFGen for generation. Furthermore, we currently train the autoencoder-based representation framework separately from the generative flow, which can be inefficient and memory-intensive.

## ETHICS STATEMENT

This work explores the core features of single-cell data and examines how capturing complex, high-dimensional cellular information can assist in answering biological questions. We aim to release CFGen as a user-friendly, open-source tool to facilitate its adoption in single-cell analysis. Given its application in biological research, CFGen may be utilized in sensitive environments that involve clinical data and patient information.

## REPRODUCIBILITY STATEMENT

Reproduction details are reported in the Appendix and the main text. The proof for Proposition 1 is extensively described in Appendix B.4, while prior knowledge on Flow Matching and classifier-free guidance is provided in Appendix B.2 and Appendix B.3. Algorithms for training and sampling with CFGen are reported in Appendix G. We introduce a thorough model description of both the autoencoder and flow components, together with modeling choices in Appendix C. Baselines and their characteristics are reported in Appendix D. All datasets are publicly available, and their source publications are referenced in the main text. We additionally summarize dataset characteristics in Table 4. Metrics and experimental setups are detailed in Appendix F.

## ACKNOWLEDGMENTS

A.P. and T.R. are supported by the Helmholtz Association under the joint research school Munich School for Data Science (MUDS). Additionally, A.P., T.R. and F.J.T. acknowledge support from the German Federal Ministry of Education and Research (BMBF) through grant numbers 031L0289A and 01IS18053A. T.R. and F.J.T. also acknowledge support from the Helmholtz Association's Initiative and Networking Fund via the CausalCellDynamics project (grant number Interlabs-0029). Additionally, F.J.T. acknowledges support from the European Union (ERC, DeepCell - grant number 101054957). Finally, A.D. acknowledges support from G-Research.

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

# A    DATASETS AND CODE

We make the code for CFGen as well as the links to pre-processed datasets available at `https://github.com/theislab/CFGen`.

# B    THEORETICAL SUPPLEMENT

## B.1    POISSON-GAMMA AND NEGATIVE BINOMIAL DISTRIBUTION

A possible parameterization of the negative binomial distribution is via a mean $\mu$ and an inverse dispersion parameter $\theta$, with the Probability Mass Function (PMF):

$$p_{\text{NB}}(x \mid \mu, \theta) = \frac{\Gamma(\theta + x)}{x!\Gamma(\theta)} \left(\frac{\theta}{\theta + \mu}\right)^{\theta} \left(\frac{\mu}{\theta + \mu}\right)^{x}. \tag{9}$$

where $\Gamma$ is the gamma function. One can show that the negative binomial distribution is obtained as a *continuous mixture* of Poisson distributions with a gamma-distributed rate. More formally, define a Poisson model $x \sim \text{Poisson}(\lambda)$ with $\lambda \geq 0$. The parameter $\lambda$ represents both the mean and variance of the distribution. Since the mean and variance are equal, the Poisson model is unsuitable for modeling over-dispersed counts (i.e., when the variance exceeds the mean). A way to overcome this limitation is to model the rate of the Poisson distribution as a random variable following a gamma distribution:

$$x \sim \text{Poisson}(\lambda), \tag{10}$$

$$\lambda \sim \text{Gamma}\left(\theta, \frac{\mu}{\theta}\right), \tag{11}$$

where $\theta > 0$ is the shape parameter and $\frac{\mu}{\theta}$ is the scale parameter. Marginalizing out $\lambda$ in the PMF of the Poisson distribution recovers the PMF of a negative binomial with mean $\mu$ and inverse dispersion $\theta$.

Crucially, the variance of this negative binomial parameterization is $\mu + \frac{\mu^2}{\theta}$. Since the additional variance term $\frac{\mu^2}{\theta}$ is always non-negative, the variance always exceeds the mean for finite $\theta$, making the negative binomial distribution well-suited for modeling over-dispersed count data.

## B.2    FLOW MATCHING WITH GAUSSIAN PATHS

Flow Matching (Lipman et al., 2023) learns a time-dependent vector field $u_t(\mathbf{z})$, where $t \in [0, 1]$, generating a probability path $p_t(\mathbf{z})$. Here, $p_0 = \mathcal{N}(\mathbf{0}, \mathbf{I})$ is the standard Gaussian prior, and $p_1$ represents a more complex target distribution. The marginal $p_t(\mathbf{z})$ is commonly formulated as a mixture:

$$p_t(\mathbf{z}) = \int p_t(\mathbf{z}|\mathbf{z}_1)q(\mathbf{z}_1)\,\mathrm{d}\mathbf{z}_1,$$

where $q$ denotes the target data distribution. The marginal vector field that generates such a mixture of paths is given by:

$$u_t(\mathbf{z}) = \int u_t(\mathbf{z}|\mathbf{z}_1)\frac{p_t(\mathbf{z}|\mathbf{z}_1)q(\mathbf{z}_1)}{p_t(\mathbf{z})}\,\mathrm{d}\mathbf{z}_1 .$$

While the marginal vector field $u_t(\mathbf{z})$ is intractable, the conditional vector field $u_t(\mathbf{z}|\mathbf{z}_1)$ has a closed-form expression when given an observed data point $\mathbf{z}_1$ and a pre-defined choice of the probability path $p_t(\mathbf{z}|\mathbf{z}_1)$. This path satisfies the boundary conditions $p_0(\cdot|\mathbf{z}_1) = p_0$ and $p_1(\cdot|\mathbf{z}_1) = \delta(\mathbf{z} - \mathbf{z}_1)$, where the Dirac delta measure $\delta(\mathbf{z} - \mathbf{z}_1)$ ensures that the distribution at time $t = 1$ is a point mass at the observed data point $\mathbf{z}_1$. Notably, regression on $u_t(\mathbf{z}|\mathbf{z}_1)$ admits the same minimizer as on $u_t(\mathbf{z})$, making it suitable for use as a target during training.

Following Lipman et al. (2023), we assume Gaussian probability paths for the transformation at each time $t$, defined by:

$$p_t(\mathbf{z}|\mathbf{z}_1) = \mathcal{N}(\mathbf{z}|\alpha_t\mathbf{z}_1, \sigma_t^2\mathbf{I}),$$

where $\alpha_t$ and $\sigma_t$ are the scheduler parameters, with $\alpha_0 = 0$, $\sigma_1 = 0$, $\alpha_1 = 1$, and $\sigma_0 = 1$. In this work, we use standard linear scheduling, where $\alpha_t = t$ and $\sigma_t = 1 - t$, linearly interpolating between the initial and final values.

### B.3 THE RELATIONSHIP BETWEEN FLOW MATCHING AND CLASSIFIER-FREE GUIDANCE (CFG)

Zheng et al. (2023) draw a relationship between CFG in score-based models (Ho & Salimans, 2021) and the Flow Matching vector field $u_t$. Specifically, the authors show that the following relationship between the score $\nabla_{\mathbf{z}} \log p_t(\mathbf{z}|y)$ and the marginal vector field $u_t(\mathbf{z}|y)$ holds:

$$u_t(\mathbf{z}|y) = a_t \mathbf{z} + b_t \nabla_{\mathbf{z}} \log p_t(\mathbf{z}|y) \,, \tag{12}$$

both in the conditional case and when $y = \varnothing$, with $a_t = \frac{\dot{\alpha}_t}{\alpha_t}$ and $b_t = (\dot{\alpha}_t \sigma_t - \alpha_t \dot{\sigma}_t) \frac{\sigma_t}{\alpha_t}$.

Let the equation

$$\nabla_{\mathbf{z}} \log \tilde{p}_t(\mathbf{z}|y) = (1 - \omega)\nabla_{\mathbf{z}} \log p_t(\mathbf{z}) + \omega \nabla_{\mathbf{z}} \log p_t(\mathbf{z}|y)$$

be the CFG score as formulated by Ho & Salimans (2021) with guidance strength $\omega$. Zheng et al. (2023) define the vector field of classifier-free Flow Matching as

$$\tilde{u}_t(\mathbf{z}|y) = (1 - \omega)u_t(\mathbf{z}) + \omega \, u_t(\mathbf{z}|y) \,. \tag{13}$$

This vector field is related to the CFG score by

$$\tilde{u}_t(\mathbf{z}|y) = a_t \mathbf{z} + b_t \nabla_{\mathbf{z}} \log \tilde{p}_t(\mathbf{z}|y) \,,$$

which is derived by substituting Eq. (12) into Eq. (13).

### B.4 PROOF OF PROPOSITION 1

**Proposition 1** *If the attributes $y_1, \ldots, y_K$ are conditionally independent given $\mathbf{z}$, the vector field*

$$\tilde{u}_t(\mathbf{z}|\mathbf{y}) = u_t(\mathbf{z}) + \sum_{i=1}^{K} \omega_i \left[u_t(\mathbf{z}|y_i) - u_t(\mathbf{z})\right]$$

*coincides with the velocity of the probability-flow ODE associated with the generative SDE of a diffusion model with the compositional score as in Eq. (7).*

*Proof. (Proposition 1)* We first justify the conditional independence assumption and subsequently prove the equality.

**Conditional independence assumption.** Given a variable $\mathbf{z}$ and a set of attributes $\mathbf{y} = y_1, ..., y_K$, we define the attribute-conditioned marginal probability path $p_t(\mathbf{z}|y_1, ..., y_K)$, with $t \in [0, 1]$. Under the assumption that the attributes are conditionally independent given $\mathbf{z}$, one can rewrite the marginal path as follows:

$$p_t(\mathbf{z}|y_1, ..., y_K) \propto p_t(\mathbf{z}, y_1, ..., y_K) = p_t(\mathbf{z}) \prod_{i=1}^{K} p_t(y_i|\mathbf{z}) \propto p_t(\mathbf{z}) \prod_{i=1}^{K} \frac{p_t(\mathbf{z}|y_i)}{p_t(\mathbf{z})} \,. \tag{14}$$

Taking the logarithm and then the gradient with respect to $\mathbf{z}$ on both sides in Eq. (14), we obtain:

$$\nabla_{\mathbf{z}} \log p_t(\mathbf{z}|y_1, ..., y_K) = \nabla_{\mathbf{z}} \log p_t(\mathbf{z}) + \sum_{i=1}^{K} \left[\nabla_{\mathbf{z}} \log p_t(\mathbf{z}|y_i) - \nabla_{\mathbf{z}} \log p_t(\mathbf{z})\right] \,. \tag{15}$$

In CFG, the goal is to sample with attribute-specific guidance strengths $\{\omega_i\}_{i=1}^{K}$ according to a modified conditional data distribution

$$\tilde{q}(\mathbf{z}|y_1, ..., y_K) \propto q(\mathbf{z}) \prod_{i=1}^{K} \left[\frac{q(\mathbf{z}|y_i)}{q(\mathbf{z})}\right]^{\omega_i} \,. \tag{16}$$

In terms of generative probabilistic paths, the score in Eq. (15) becomes:

$$\nabla_{\mathbf{z}} \log \tilde{p}_t(\mathbf{z}|y_1, ..., y_K) = \nabla_{\mathbf{z}} \log p_t(\mathbf{z}) + \sum_{i=1}^{K} \omega_i \left[\nabla_{\mathbf{z}} \log p_t(\mathbf{z}|y_i) - \nabla_{\mathbf{z}} \log p_t(\mathbf{z})\right] \,. \tag{17}$$

In score-based models, the formulation in Eq. (17) is used to parameterize the drift of the reverse-time SDE that generates data points conditionally on the attributes $y_1, ..., y_K$ with guidance strengths $\{\omega_i\}_{i=1}^{K}$.

**Proof of equality.** Following the standard theory of score-based models (Yang et al., 2023) and their compositional extension (Liu et al., 2022), we first note that one can use the compositional CFG score

$$\nabla_{\mathbf{z}} \log \tilde{p}_t(\mathbf{z}|\mathbf{y}) = \nabla_{\mathbf{z}} \log p_t(\mathbf{z}) + \sum_{i=1}^{K} \omega_i \left[ \nabla_{\mathbf{z}} \log p_t(\mathbf{z}|y_i) - \nabla_{\mathbf{z}} \log p_t(\mathbf{z}) \right] . \tag{18}$$

to simulate the generative probability-flow ODE:

$$\dot{\mathbf{z}} = f_t \mathbf{z} - \frac{1}{2} g_t^2 \nabla_{\mathbf{z}} \log \tilde{p}_t(\mathbf{z}|\mathbf{y}) . \tag{19}$$

Given a scheduling pair $(\alpha_t, \sigma_t)$, one can show (Kingma et al., 2021) that the drift and diffusion coefficients of a score-based diffusion model following the formulation from Song et al. (2021) satisfy

$$f_t = \frac{d \log \alpha_t}{dt}, \quad g_t^2 = \frac{d\sigma_t^2}{dt} - 2 \frac{d \log \alpha_t}{dt} \sigma_t^2 . \tag{20}$$

From these, we derive

$$a_t = \frac{\dot{\alpha}_t}{\alpha_t}, \quad b_t = (\dot{\alpha}_t \sigma_t - \alpha_t \dot{\sigma}_t) \frac{\sigma_t}{\alpha_t} . \tag{21}$$

Thus, rewriting Eq. (19), we obtain the probability-flow ODE in the form:

$$\dot{\mathbf{z}} = a_t \mathbf{z} + b_t \nabla_{\mathbf{z}} \log \tilde{p}_t(\mathbf{z}|\mathbf{y}) . \tag{22}$$

Next, using results from Zheng et al. (2023), as discussed in Appendix B.3, we write the unconditional and conditional vector fields as

$$u_t(\mathbf{z}) = a_t \mathbf{z} + b_t \nabla_{\mathbf{z}} \log p_t(\mathbf{z}) , \tag{23}$$

$$u_t(\mathbf{z}|y_i) = a_t \mathbf{z} + b_t \nabla_{\mathbf{z}} \log p_t(\mathbf{z}|y_i) . \tag{24}$$

Plugging Eqs. (23) and (24) into the Eq. (8) for the compositional flow, we obtain

$$\tilde{u}_t(\mathbf{z}|\mathbf{y}) = u_t(\mathbf{z}) + \sum_{i=1}^{K} \omega_i \left[ u_t(\mathbf{z}|y_i) - u_t(\mathbf{z}) \right] \tag{25}$$

$$= a_t \mathbf{z} + b_t \left[ \nabla_{\mathbf{z}} \log p_t(\mathbf{z}) + \sum_{i=1}^{K} \omega_i \left( \nabla_{\mathbf{z}} \log p_t(\mathbf{z}|y_i) - \nabla_{\mathbf{z}} \log p_t(\mathbf{z}) \right) \right] \tag{26}$$

$$= a_t \mathbf{z} + b_t \nabla_{\mathbf{z}} \log \tilde{p}_t(\mathbf{z}|\mathbf{y}) . \tag{27}$$

This matches Eq. (22), completing the proof.

## B.5 RELATIONSHIP WITH EXISTING SINGLE-CELL GENERATIVE MODELS

Although likelihood-based models are standard in the single-cell literature, CFGen leverages a novel factorization scheme, as depicted in Eq. (5). Below, we outline key differences between our approach and standard single-cell VAEs:

- In scVI (Lopez et al., 2018), the conditioning is applied during the decoding phase rather than at the prior level for the latent variable $p(\mathbf{z})$. In contrast, CFGen conditions directly on the latent prior $p(\mathbf{z}|y)$, allowing it to sample from multiple modes. This enables a more flexible generation compared to traditional approaches that rely solely on decoder-based conditioning.

- Consequently, the likelihood term $p(\mathbf{x}|\mathbf{z}, l)$ differs from that of most single-cell VAEs. In CFGen, this term is modeled using an unconditional decoder since the conditioning on $y$ is already incorporated in the flow-based generation of $\mathbf{z}$. Conversely, standard conditional VAEs must explicitly feed the label into both the encoder and decoder.

- While some VAEs incorporate a conditional prior (Xu et al., 2021), they are primarily designed for representation learning rather than generative modeling. These models often under-regularize the latent space to favor structure and reconstruction. In contrast, CFGen enforces a *strong* conditional flow-based prior on the latent cell representation, avoiding the trade-off between Kullback-Leibler divergence minimization and likelihood optimization. As a result, it can generate high-quality samples without requiring a highly regularized latent space.

- Although previous works have explored defining a distribution over library size, our approach uniquely integrates it into the generative process through our specific factorization in Eq. (5). Typically, the library size is used merely as a scaling factor in likelihood optimization, applied to the post-softmax output of the decoder. CFGen instead employs it as a conditioning variable for sampling from the flow-based conditional prior, ensuring a formally sound integration. Specifically, we define the library size as a conditioning attribute and factorize our latent variable model to generate $\mathbf{z}$ from $p(\mathbf{z}|y, l)$. To the best of our knowledge, this formulation has not been explored before.

### B.6 Conditioning on the size factor

Single-cell VAEs, like scVI, offer the option to learn a log-normal distribution over the size factor, which can then be sampled. This process is similar to our approach, as both methods fit a log-normal distribution over the library size in the data. However, the key distinction lies in how the size factor interacts with the latent cellular state.

In scVI, the latent cellular state and the size factor are sampled independently. In contrast, CFGen provides the option to bias the sampling by the size factor. More specifically, CFGen samples a latent state that inherently accounts for cell size, whereas scVI does not.

If the size factor is relatively uniform within a target population of cells, the approach in scVI might be sufficient. However, if we generate conditioned on a coarse annotation (e.g., the source study of a dataset in modern atlases), the library size can vary significantly within the annotation category (see Fig. A17 for examples using different studies in the Human Cell Atlas dataset). Sampling a latent state and size factor independently in such cases may lead to inconsistencies—scaling a decoded cell state by an incompatible size factor could produce unrealistic results.

Instead, conditioning on the size factor is more appropriate, as it biases the sampling of the latent state toward regions of the latent space where that specific size factor is naturally represented. This strategy can be combined with coarsely annotated variables to enable more targeted conditional generation.

## C Model details

The CFGen model is implemented in `PyTorch` (Paszke et al., 2017), version `2.1.2`.

### C.1 The CFGen Autoencoder

Before training the flow model to generate noise from data, we first embed the data using an autoencoder trained with maximum likelihood optimization.

**Encoder.** The encoder is a multi-layer perceptron (MLP) with two hidden layers of dimensions `[512, 256]`. The final layer maps the input to a latent space, whose dimensionality is dataset-dependent. In our experiments, we use a 50-dimensional latent space for most datasets, except for the Human Lung Cell Atlas (HLCA) and Tabula Muris, where we set the latent space to 100 dimensions for increased representational capacity. In the multi-modal setting, different data modalities are embedded into a shared latent space. Each modality is first processed by a modality-specific MLP encoder. Due to the high dimensionality of DNA accessibility data (referred to as ATAC data from Assay for Transposase-accessible Chromatin), its encoder uses hidden layers of dimensions `[1024, 512]`. The outputs of the RNA and ATAC encoders are then concatenated and passed through a shared encoder layer, mapping to a final 100-dimensional latent representation.

**Decoder.** The decoder maps the latent space to the parameter space of a likelihood model. For multi-modal data, each modality has its dedicated decoder.

- scRNA-seq: The latent representation is mapped to the mean parameter $\boldsymbol{\mu}$ of a negative binomial likelihood, with one dimension per gene. Following Lopez et al. (2018), we apply a `softmax` transformation across genes to produce normalized probabilities. These probabilities are then scaled by the library size (total transcript count per cell). The inverse dispersion parameter of the negative binomial distribution is a learned model parameter, implemented

via `torch.nn.Parameter`. We offer the option to model inverse dispersion per gene or gene-attribute pair, depending on dataset properties.

- ATAC-seq: The decoder maps the latent space to continuous logit values. These are passed through an elementwise `sigmoid` function to produce probabilities per genomic region. Unlike RNA data, no size factor scaling is required.

**Additional training details.** We train the encoder and decoder networks jointly via likelihood optimization. Therefore, the weights of the networks are optimized (along with the inverse dispersion parameter) to produce the parameters that maximize the likelihood of the data under a predefined noise model. For scRNA-seq, we employ a negative binomial distribution, while for ATAC-seq, we use a Bernoulli likelihood. The losses from different modalities are summed before applying backpropagation. Note that scRNA-seq data are provided by the encoder in their $\log$-transformed version for training stability. However, the loss is evaluated on the original count data.

For all settings, we set the learning rate to $0.001$, with all layer pairs interleaved with one-dimensional batch normalization layers. We use the `AdamW` optimizer and the `ELU` activation function as the non-linearity.

## C.2 THE VELOCITY MODEL

**The vector field architecture.** The vector field model takes as input the *latent representation* computed by the encoder and produces a vector field used to simulate paths that generate data from noise. The architecture follows a deep ResNet (He et al., 2016) whose output has the same number of features as the input. The velocity model consists of the following components:

- A linear projection layer that maps the input dimension to the hidden dimension of the flow model.
- Three stacked ResNet blocks are responsible for representation learning and conditioning.
- An output layer with a single non-linearity, implemented using a `SiLU` activation function.
- A time embedder that encodes time using sinusoidal multi-dimensional embeddings (Vaswani et al., 2017) with a frequency value of `1e4`. This embedder is an MLP with two layers and `SiLU` non-linearity.
- A size factor embedder that applies sinusoidal embeddings (Vaswani et al., 2017) to guide generation towards a predefined number of transcripts. This component is used only when the size factor is a conditioning variable, i.e., when we do not assume $p(\mathbf{z}|y,l) = p(\mathbf{z}|y)$. Before being passed through the sinusoidal embeddings, the $\log$ size factor is normalized to a range approximately between $0$ and $1$, using the maximum and minimum $\log$ size factors in the dataset.
- Covariate embeddings for all different conditioning attributes.

**Additional technical details.** During training, the covariate embeddings are summed elementwise with the time embedding and, if applicable, the size factor embedding. Thus, all embeddings are either designed to have the same dimensionality or are transformed to a common size. This summed representation is then passed as a single vector to the ResNet blocks.

Both the summed conditioning embedding and the down-projected input are provided to the ResNet blocks, which consist of:

- A non-linear input transformation of the state embedding.
- A linear encoder for the covariate embedding.
- A non-linear output transformation.
- A skip connection.

The outputs of the non-linear input transformation and the covariate encoder are summed and passed through the output transformation. The result is then added to the input of the ResNet via the skip connection, following the traditional residual block structure (He et al., 2016). All non-linear transformations are implemented as simple `[SiLU, Linear]` stacks.

In the standard setting, we train the flow model for $1,000$ epochs using the `AdamW` optimizer, a learning rate of $0.001$, and a batch size of $256$.

### C.3 COVARIATE EMBEDDINGS

Covariate embeddings are trainable `torch.nn.Embdding` layers of pre-defined size. In our experiments, we use a size of $100$ in most of the settings.

### C.4 SAMPLING FROM NOISE

To generate discrete observations from noise, we first draw a covariate from the associated categorical distribution, with proportions estimated from the observed data. Next, we sample a size factor from a LogNormal distribution, where the mean and standard deviation are set as the Maximum Likelihood Estimates (MLE) from the entire dataset or conditioned on a technical effect covariate. We then sample Gaussian noise and simulate a latent observation from the real dataset conditionally. This is done by integrating the vector field computed by the neural network in Appendix C.2, starting from Gaussian noise. The integration is performed using the `dopri5` solver with `adjoint` sensitivity and a tolerance of `1e-5` from the `torchdyn` package (Poli et al., 2021) in Python3 (Van Rossum & Drake, 2009), over the $[0, 1]$ time interval. The generated latent vector is then decoded into the parameter space of the noise model for the data—negative binomial for scRNA-seq or Bernoulli for ATAC-seq—after which single cells are sampled.

### C.5 SEPARATE TRAINING

In CFGen, we train the encoder $f_\eta$ separately from the flow model. Initially, when we attempted to model the autoencoder and the flow jointly, we found that training the flow was unstable. Specifically, Flow Matching performs better when the state space is fixed. Alternating between autoencoder and flow updates leads to continuous changes in the data representation, as the autoencoder evolves with the flow. This dynamic hinders accurate velocity field estimation, particularly during the early updates of the autoencoder. One could initially train the autoencoder with a higher learning rate than the flow, periodically decreasing the former and increasing the latter. However, this approach is essentially similar to training the autoencoder and flow separately, which is the strategy we ultimately adopt to avoid the need for repeatedly retraining the autoencoder.

### C.6 SCHEDULING

We use linear scheduling, following the original formulation from Lipman et al. (2023).

## D BASELINE DESCRIPTION

### D.1 SCVI, MULTIVI, PEAKVI

scVI (Lopez et al., 2018), MultiVI (Ashuach et al., 2023), and PeakVI (Ashuach et al., 2022) are all VAE-based generative models designed for single-cell discrete data. Following the standard VAE framework, these models learn a Gaussian latent space, which is then decoded into the parameters of discrete likelihood models that describe different single-cell modalities. While scVI and PeakVI generate single modalities—scVI for scRNA-seq and PeakVI for ATAC data—MultiVI learns a shared latent space across modalities while maintaining separate discrete decoders for each data type.

### D.2 SCANVI AND SCPOLI

In the batch correction experiment described in Section 5.5, we compare CFGen with two additional VAE-based models: scANVI (Xu et al., 2021) and scPoli (De Donno et al., 2023). scANVI extends scVI by incorporating a latent cell type classifier to preserve biological structure in the representation space and introducing a conditional prior on the latent space. scPoli differs from scANVI by using continuous embeddings instead of one-hot encodings as conditioning inputs to the VAE. Additionally, it enforces biological coherence by aligning cellular representations with latent cell-type prototypes.

In simple terms, scPoli encourages cells to cluster around the average embedding vector of their respective cell types.

### D.3   SCGAN

The scGAN model (Marouf et al., 2020) is a Generative Adversarial Network (GAN) (Goodfellow et al., 2014) designed for realistic scRNA-seq data generation. It minimizes the Wasserstein distance between real and generated cell distributions, employing a generator network to produce synthetic samples and a critic network to distinguish real from generated cells. The architecture includes fully connected layers and a custom library-size normalization (LSN) layer for stable training. The model extends to conditional scGAN (cscGAN) for type-specific cell generation. Evaluation relies on metrics such as t-SNE visualization and marker gene correlation to assess the quality of generated cells. While scGAN has been explored for conditional generation, we found that conditioning on cell type led to significantly worse results compared to an alternative approach in which the model is conditioned on data-driven Leiden cluster labels rather than real cell-type labels. We refer to this alternative as the *unconditional* version, as it does not use predefined labels but instead leverages cluster-derived attributes.

### D.4   SCDIFFUSION

scDiffusion (Luo et al., 2024) is a generative model that leverages diffusion models to generate realistic single-cell gene expression data. The model consists of three main components:

- **An autoencoder**, which maps gene expression profiles to a latent space, enabling compression and feature extraction from high-dimensional single-cell data.
- **A diffusion backbone network**, which learns to reverse a diffusion process applied to the latent embeddings, progressively refining noisy representations into meaningful biological signals.
- **A conditional classifier**, which guides the generative process by incorporating cell type or other biological attributes, ensuring controlled cell generation.

During training, the autoencoder first encodes real single-cell data into a latent representation. Noise is then progressively added to these embeddings following a predefined diffusion schedule. The diffusion backbone network is trained to learn the reverse process, reconstructing clean embeddings from noisy ones. Simultaneously, the conditional classifier is optimized to predict labels from these latent representations, reinforcing biological relevance in the learned distribution.

At inference, scDiffusion starts from a random noise vector in the latent space and iteratively removes noise using the trained diffusion backbone, ultimately generating a clean embedding. This embedding is then decoded by the autoencoder to reconstruct synthetic gene expression data. The process enables controlled single-cell generation by conditioning on specific biological attributes.

### D.5   DISCUSSION: KEY DIFFERENCES BETWEEN SCDIFFUSION AND CFGEN

**Handling Single-Cell Data Properties.**   scDiffusion applies Gaussian diffusion to preprocessed single-cell data, disregarding key properties such as sparsity, overdispersion, and discreteness. While normalization ensures data continuity, most single-cell methods preserve zeros and non-linear mean-variance trends. Since continuous decoders like scDiffusion require centered, dense inputs, their design is suboptimal for scRNA-seq data.

**Conditional Sampling.**   scDiffusion relies on classifier-based guidance, making conditional generation highly dependent on classifier accuracy. This limits its ability to generate rare cell types or handle attributes that are difficult to classify.

**Efficiency and Sampling Speed.**   CFGen is two to three orders of magnitude faster than scDiffusion due to:

- Efficient Sampling: Flow Matching directly maps noise to data along nearly straight paths, requiring only 5–10 integration steps compared to the >1000 steps needed for scDiffusion.

Table 4: List of datasets considered in this work with the associated number of genes, cells, and cell types.

| Dataset name | Number of cells | Number of genes | Number of cell types |
|---|---|---|---|
| PMBC3K | 2,638 | 8,573 | 8 |
| Dentate gyrus | 18,213 | 17,002 | 14 |
| Tabula Muris | 245,389 | 19,734 | 123 |
| HLCA | 584,944 | 27,997 | 50 |
| PBMC10k | 10,025 | 25,604 | 14 |
| NeurIPS | 90,261 | 14,087 | 45 |
| PBMC COVID | 422,220 | 2,000 | 29 |
| C.Elegans | 89,701 | 17,747 | 35 (plus unknown) |

- Lower Dimensionality: CFGen operates in a compact latent space (50–100 dimensions), whereas scDiffusion performs denoising in a much higher-dimensional space (1000 dimensions).

- Guidance approach: Unlike scDiffusion, which relies on a classifier's gradient, CFGen uses CFG-based guidance, avoiding performance bottlenecks due to classifier accuracy.

## E    DATA PREPROCESSING AND DESCRIPTION

Single-cell data were preprocessed using `scanpy` (Wolf et al., 2018). Count normalization was applied only to baseline models requiring real-valued inputs. In these cases, gene counts were library-size normalized to `1e4` and `log`-transformed. Since CFGen, MultiVI, PeakVI, and scVI operate in discrete space, we trained them on raw counts without normalization. Additionally, we filtered out genes expressed in fewer than 20 cells across all datasets.

## F    EXPERIMENT DESCRIPTION AND EVALUATION METRICS

### F.1    2-WASSERSTEIN DISTANCE AND MMD

We use the 2-Wasserstein distance and the RBF-kernel Mean Maximum Discrepancy (MMD) with scales $\{0.01, 0.1, 1, 10, 100\}$ (Gretton et al., 2012) to measure the overlap between real and generated data. To implement the former, we use the Python Optimal Transport (POT) (Flamary et al., 2021) package. For the MMD, we resort to the implementation proposed in [3].

### F.2    DISTRIBUTION METRICS COMPARISONS

To compute the metrics in Table 1, we generate three datasets per model, each matching the size of the original. In the conditional setting, we compute distribution metrics per cell type, comparing subsets of real and generated data. In the unconditional setting, we sample batches of $5,000$ cells from the full distribution. To mitigate the curse of dimensionality, we compute MMD and 2-Wasserstein distances in a 30-dimensional Principal Component (PC) space. Generated cells are projected using PC loadings from real cells to ensure comparability. Since scDiffusion and scGAN generate normalized data while CFGen and scVI produce discrete counts, we normalize CFGen and scVI outputs to a total of `1e4` counts per cell and then apply a `log`-transformation. The same processing is applied to real data, ensuring all models operate on comparable quantities. All metrics are reported on the test set.

### F.3    VARIANCE-MEAN TREND AND SPARSITY HISTOGRAMS

After pre-processing (as described in Appendix F.2), we compute the mean and variance of gene expression across cells and the frequency of unexpressed genes per cell. The mean-variance trend in raw count data is expected to be quadratic (see Appendix B.1). However, normalization and `log`-transformation—required for comparison with scDiffusion and scGAN—alter this trend. Despite this, examining the empirical mean-variance relationship remains informative, as it should align with real data behavior.

---

[3] https://github.com/atong01/conditional-flow-matching

### F.4 MULTI-MODAL EVALUATION

We generate multi-modal data and perform an unconditional comparison with the ground truth, as described in Appendix F.2. For ATAC data, we normalize both real and generated cells using the TF-IDF algorithm from the MUON package (Bredikhin et al., 2022).

To generate Fig. A7b, we follow these steps:

1. Compute the average gene expression and peak accessibility (i.e., fraction of accessible regions) per cell type for marker genes/peaks, following [4]. This yields a `cell_type x marker` matrix for both real and generated datasets, where each entry represents the mean expression or accessibility of a marker in a given cell type.

2. Correlate the row vectors of these matrices between real and generated datasets. A high correlation indicates that the generative model accurately captures mean marker expression and accessibility per cell type.

### F.5 GUIDANCE STRENGTH EXPERIMENTS

Fig. 3 illustrates the qualitative performance of guidance in CFGen. First, we train CFGen on each dataset following Algorithm 1. Upon successful training, we sample 500 cells under different guidance strength combinations, varying one attribute while keeping the other fixed, as shown in the figure. For unconditional generation (i.e., guidance strength of 0 for both attributes), we generate as many cells as in the real dataset to better visualize the overlap between real and synthetic data. When applying guidance, we train the guided CFGen model with an unconditional sampling probability of $p_{\text{uncond}} = 0.2$ (see Algorithm 1).

### F.6 SCGPT (CUI ET AL., 2024) GENERALIZATION PERFORMANCE ENHANCEMENT

We split the PBMC COVID and HLCA datasets into a training set and a held-out set, ensuring a more challenging generalization task by leaving out all cells from 20% of the donors in both datasets. This results in 80 training and 27 test donors for HLCA and 60 training and 15 test donors for PBMC COVID. After augmenting the training set (see Section 5.4), we use a pre-trained scGPT model to embed both the training and validation sets for both the original and augmented data. We then fit a k-Nearest-Neighbor (kNN) classifier on the training embeddings and evaluate its performance on the held-out set. The results, shown in Fig. 4, illustrate how the classification performance in terms of recall on a cell type varies as a function of its frequency in the dataset after augmentation. A performance improvement suggests that the additional synthetic examples help the model better distinguish the cell type in the LLM representation space.

### F.7 BATCH CORRECTION EVALUATION

**Correction with CFGen.** Batch correction aims to remove technical effects while preserving biological variation in single-cell data. Given an observation $\mathbf{x}$ associated with a batch label $y_{\text{batch}}$ and a biological annotation $y_{\text{cell type}}$, we first encode $\mathbf{x}$ into a latent variable $\mathbf{z}$.

Flow Matching is a generative model that maps a prior distribution to the data distribution via an invertible flow. This invertibility allows data distributions to be transported to noise. Inverting the flow back to the prior removes batch effects as well as cell-type variability from $\mathbf{z}$ (Rombach et al., 2020).

To perform batch correction, we first apply flow inversion to remove both biological and technical variation from $\mathbf{z}$. Then, we simulate the flow forward again, starting from the noise representation. Given a reference batch $y_{\text{batch}}^{\text{ref}}$, we generate new observations conditioned on this batch while preserving the original biological label $y_{\text{cell type}}$. When applied across the dataset, this procedure aligns all observations to the same batch, effectively removing batch-specific variations.

---

[4] https://muon-tutorials.readthedocs.io/en/latest/single-cell-rna-atac/pbmc10k/3-Multimodal-Omics-Data-Integration.html

**Remarks.** In the context of CFG, the weights $\omega_{\text{batch}}$ and $\omega_{\text{cell type}}$ control the degree of biological preservation. Notably, our batch correction approach is conceptually similar to style transfer methods in diffusion models (Wang et al., 2023).

**Evaluation Setup.** We train CFGen and all competing models using identical cell type and batch covariates. For a fair comparison, we use a latent space of 50 dimensions for all models. For uncorrected data, batch mixing is evaluated in the PC space.

All VAE-based models were trained for 100 epochs with default settings, while scPoli was pre-trained for 40 steps, following De Donno et al. (2023).

**Metrics.** To assess batch correction and biological conservation, we use the scIB package (Luecken et al., 2021b). Specifically, we employ five metrics for batch correction and five for biological conservation. The scores reported in Table 3 correspond to the average of these metrics, as computed by scIB. All scores are normalized between 0 and 1, where 1 indicates perfect correction or conservation.

All metrics rely on kNN graphs, using batch and cell-type labels to evaluate technical and biological mixing. Below, we briefly describe each metric, though we refer to Luecken et al. (2021b) for further details.

1. **Batch Correction Metrics:**
   - Silhouette Batch – Measures the Average Silhouette Width (ASW) between batch clusters.
   - iLISI – Computes the Inverse Simpson's Index based on neighborhood composition in kNN graphs, indicating batch mixing quality.
   - KBET – Evaluates whether the local batch composition in a cell's kNN neighborhood matches the expected global batch distribution.
   - Graph Connectivity – Assesses whether cells sharing the same label form a fully connected subgraph in the kNN representation.
   - PCR (Principal Component Regression) – Quantifies batch-associated variance before and after correction.

2. **Biological Conservation Metrics:**
   - Isolated Labels – Identifies rare cell types appearing in the fewest number of batches and assesses their separation from other cell identities.
   - K-means NMI – Computes the Normalized Mutual Information (NMI) between k-means clustering and batch clusters.
   - K-means ARI – Measures the Adjusted Rand Index (ARI) between k-means clustering and batch clusters.
   - Silhouette Label – Represents the Average Silhouette Width (ASW) between cell type clusters.
   - cLISI – A cell-type-specific version of the iLISI score, evaluating biological structure preservation.

**Selection of the guidance weights for batch correction.** Appendix H.8 provides an intuition for the selection process. In batch correction, cells are transported to noise and back to data guided by biological and batch covariates. The guidance strength parameters $\omega_{\text{bio}}$ and $\omega_{\text{batch}}$ determine the emphasis on biological conservation and batch correction. Based on the scIB metrics only, one might select the highest guidance strengths, as these maximize aggregation within cell types and batches. However, as shown in Fig. A15 and Fig. A16, scIB metrics alone can be misleading and should be paired with qualitative evaluation. Excessive guidance collapses variability beyond the batch and biological annotations, leading to artefacts. Parameters near $\omega_{\text{bio}}, \omega_{\text{batch}} \in \{1, 2\}$ generally balance signal preservation and correction effectively. For example, Fig. A16 demonstrates that excessive biological preservation causes unnatural clustering. The extent of the batch effect in the data should also guide parameter selection. For C. Elegans, with mild batch effects, $\omega_{\text{bio}} = 2, \omega_{\text{batch}} = 1$ performs better than $\omega_{\text{bio}} = 1, \omega_{\text{batch}} = 2$. Conversely, for NeurIPS, $\omega_{\text{bio}} = 1, \omega_{\text{batch}} = 2$ avoids artifacts observed for $\omega_{\text{bio}} > 1$ (Fig. A16). In summary, we recommend assessing the batch effect severity, sweeping over guidance weights, and selecting parameters that optimize scIB metrics without compromising realistic single-cell representations.

## G  ALGORITHMS

Algorithm 1 and Algorithm 2 depict our training strategies. In what follows, for notational simplicity, we indicate $\phi_t(\mathbf{z})$ with $\mathbf{z}_t$, where $t \in [0, 1]$.

---

**Algorithm 1** Train CFGen with multiple attributes on scRNA-seq

---

**Require:** Probability of unconditional generation $p_{\text{uncond}}$, trained encoder $f_\eta$, scheduling $(\alpha_t, \sigma_t)$.
1: Initialize $v_{t,\xi}$
2: **while** not converged **do**
3:     Sample $(\mathbf{x}, y_1, ..., y_K)$ from the data
4:     $\mathbf{z}_1 \leftarrow f_\eta(\mathbf{x})$
5:     Sample $t$ from $\mathcal{U}[0, 1]$
6:     $l \leftarrow$ Sum of entries of $\mathbf{x}$
7:     Sample $b$ from $\text{Bernoulli}(p_{\text{uncond}})$
8:     **if** $b = 1$ **then**
9:         $y \leftarrow \emptyset$
10:     **else**
11:         $y \leftarrow$ sample uniformly a label among $y_1, ..., y_K$
12:     **end if**
13:     $\mathbf{z}_0 \sim \mathcal{N}(\mathbf{0}, \mathbf{I})$ {sample noise}
14:     $\mathbf{z}_t \leftarrow \alpha_t \mathbf{z}_1 + \sigma_t \mathbf{z}_0$ {noisy data point}
15:     $\dot{\mathbf{z}}_t \leftarrow \dot{\alpha}_t \mathbf{z}_1 + \dot{\sigma}_t \mathbf{z}_0$
16:     Take gradient step on $\nabla_\xi ||v_{t,\xi}(\mathbf{z}_t, y, l) - \dot{\mathbf{z}}_t||^2$
17: **end while**
**Output:** $v_{t,\xi}$

---

---

**Algorithm 2** Sampling from multi-attribute guided CFGen for scRNA-seq

---

**Require:** Trained velocity field $v_{t,\xi}$, conditions $y_1, ..., y_K$, guidance parameters $\omega_1, ..., \omega_K$,
    size factor distribution parameters $(\mu_l, \sigma_l)$, number of ODE steps $n_{\text{ode}}$,
    trained decoder $h_\psi$, trained inverse dispersion parameter $\boldsymbol{\theta}$.
1: Sample size factor $l$ from $\text{LogNormal}(\mu_l, \sigma_l)$
2: $\mathbf{z}_0 \sim \mathcal{N}(\mathbf{0}, \mathbf{I})$ {sample noise}
3: $n \leftarrow 1/n_{\text{ode}}$ {step size}
4: $\tilde{u}_t(\cdot) \leftarrow v_{t,\xi}(\cdot, \emptyset, l) + \sum_{i=1}^{K} \omega_i [v_{t,\xi}(\cdot, y_i, l) - v_{t,\xi}(\cdot, \emptyset, l)]$ {guided velocity function}
5: **for** $t = 0, n, ..., 1 - n$ **do**
6:     $\mathbf{z}_{t+n} \leftarrow \text{ODEStep}(\tilde{u}_t, \mathbf{z}_t)$ {ODE solver step}
7: **end for**
8: $\mathbf{x} \leftarrow$ Sample from $\text{NB}(l \, \text{softmax}(h_\psi(\mathbf{z}_1)), \boldsymbol{\theta})$
**Output:** $\mathbf{x}$

---

# H ADDITIONAL RESULTS

## H.1 ANALYSIS OF THE RUNTIME

In Fig. A1, we empirically evaluate how different hyperparameters impact CFGen's runtime. To do so, we generate synthetic data using an untrained CFGen instance initialized with a specific configuration and run our experiments on an NVIDIA A100 GPU. Each hyperparameter is assessed across different latent space sizes, as the latent space dimension serves as a bottleneck in Flow Matching models and is expected to have the greatest influence on generation speed.

We consider the following hyperparameters:

1. Number of generated genes (default: 20,000).

2. Number of generated cells (default: 50,000).

3. Latent space dimensionality of the denoising model's bottleneck (default: 128).

4. Number of neural network blocks in the denoising model (default: 3).

5. Embedding size for conditional inputs (default: 128).

When evaluating the effect of a specific hyperparameter, all others are fixed at their default values to ensure a controlled comparison.

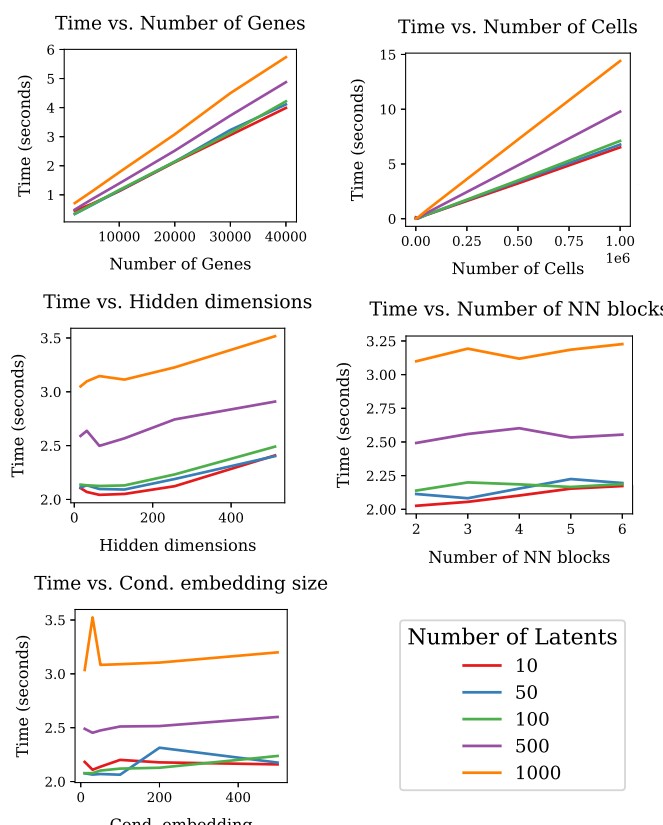

Figure A1: Runtime analysis of the CFGen generation process. Each panel represents a different hyperparameter configuration. Different lines in each plot correspond to varying latent space dimensionalities, which directly impact generation time. We examine how runtime changes as a function of five key hyperparameters: (1) number of generated genes, (2) number of generated cells, (3) hidden dimension of the denoising model, (4) number of blocks in the denoising model, and (5) size of the condition embedding. Results are reported in seconds. When varying one hyperparameter, all others are fixed at their default values.

In Fig. A1, we observe that the most influential hyperparameters affecting generation speed are the number of cells and genes, while factors related to the neural network size have a smaller impact. As expected, the number of latent codes significantly influences the sampling speed, as it determines the dimensionality of the generation space. Additionally, we compare the training and sampling runtimes of CFGen against competing models across different datasets (see Table 5 and Table 6).

Table 5: Training runtime table. Each entry corresponds to the time in seconds required to train a model on different datasets. The number of cells and genes composing each dataset is reported at the bottom of the table. CFGen and scDiffusion are broken down into their different components that should be considered additively for an overview of the total runtime. For all the models, the batch size is set to 128.

|  | PBMC3K | Dentate gyrus | Tabula muris | HLCA | PBMC10K (scRNA-seq) |
|---|---|---|---|---|---|
| CFGen Flow | 1.02 | 7.13 | 69.00 | 192.12 | 3.23 |
| CFGen AE | 1.40 | 6.31 | 68.40 | 253.21 | 6.30 |
| scVI | 0.08 | 2.11 | 18.13 | 65.12 | 2.15 |
| MultiVI | - | - | - | - | 22.12 |
| scDiffusion Denoiser | 1.03 | 2.13 | 18.02 | 53.62 | 4.48 |
| scDiffusion AE | 0.98 | 7.14 | 165.6 | 329.02 | 7.39 |
| scDiffusion classifier | 0.01 | 0.71 | 9.66 | 26.32 | 0.39 |
| scGAN | 0.98 | 5.15 | 20.41 | 181.12 | 2.40 |
| **No. of cells** | 2,638 | 18,213 | 245,389 | 584,944 | 10,025 |
| **No. of genes** | 8,573 | 17,002 | 19,734 | 27,997 | 25,604 |

Table 6: Generation runtime table. Each entry corresponds to the time in seconds required for a model to generate as many cells and genes as in the original dataset. The number of cells and genes composing each dataset is reported at the bottom of the table.

|  | PBMC3K | Dentate gyrus | Tabula muris | HLCA | PBMC10K (scRNA-seq) |
|---|---|---|---|---|---|
| CFGen | 0.34 | 0.26 | 3.68 | 8.62 | 0.43 |
| scVI | 0.01 | 0.02 | 1.26 | 3.63 | 0.03 |
| MultiVI | - | - | - | - | 0.03 |
| scDiffusion | 48.79 | 105.08 | 1255.41 | 2004.00 | 113.41 |
| scGAN | 0.70 | 0.94 | 4.15 | 12.39 | 0.68 |
| **No. of cells** | 2,638 | 18,213 | 245,389 | 584,944 | 10,025 |
| **No. of genes** | 8,573 | 17,002 | 19,734 | 27,997 | 25,604 |

From the sampling runtime results in Table 6, we observe that VAE-based models (scVI and MultiVI) are generally faster. However, it is important to note that these models are inherently less expressive and perform worse than CFGen, particularly on large datasets (see Table 1 and Fig. A6). Notably, CFGen outperforms scDiffusion in speed, accelerating sampling by orders of magnitude. This improvement is attributed to the following factors:

- CFGen requires fewer simulation steps than scDiffusion (5-10 steps in CFGen vs. >1000 in scDiffusion) while achieving superior empirical and quantitative results.

- CFGen operates in a lower-dimensional latent space (50-100 dimensions vs. 1000 dimensions for scDiffusion, as recommended in the manuscript).

- CFGen employs classifier-free guidance, whereas scDiffusion relies on classifier-based guidance. Consequently, CFGen's performance is not affected by the gradient of a classifier's prediction at each step.

Strikingly, CFGen can generate comprehensive atlases with over 500,000 cells, such as HLCA, in just 8 seconds. For fairness, we acknowledge that the speedup depends on the batch size that can fit into memory during sampling (10k cells in our case).

## H.2 EXAMPLE OF SYNTHETIC GENERATION BY CFGEN

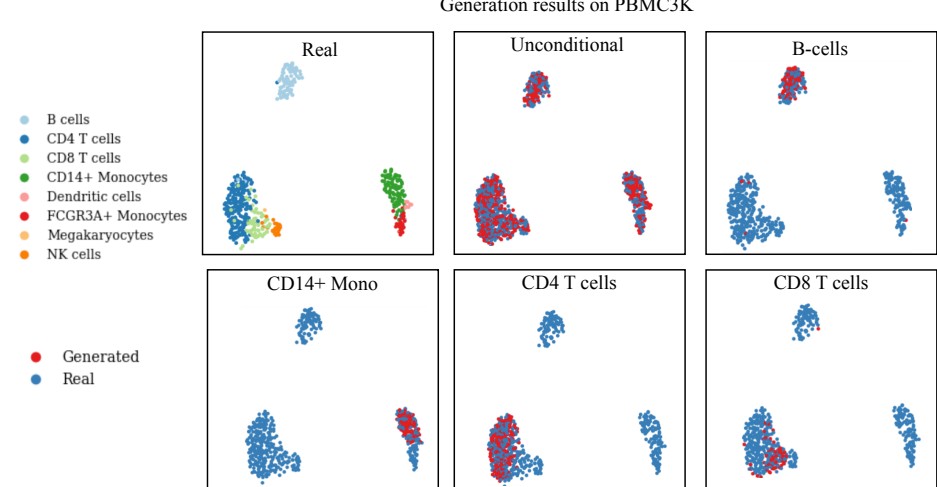

Figure A2: Uni-modal generation of scRNA-seq by CFGen on the PBMC3K dataset. Real and generated cells are embedded together and visualized as 2D UMAP coordinates.

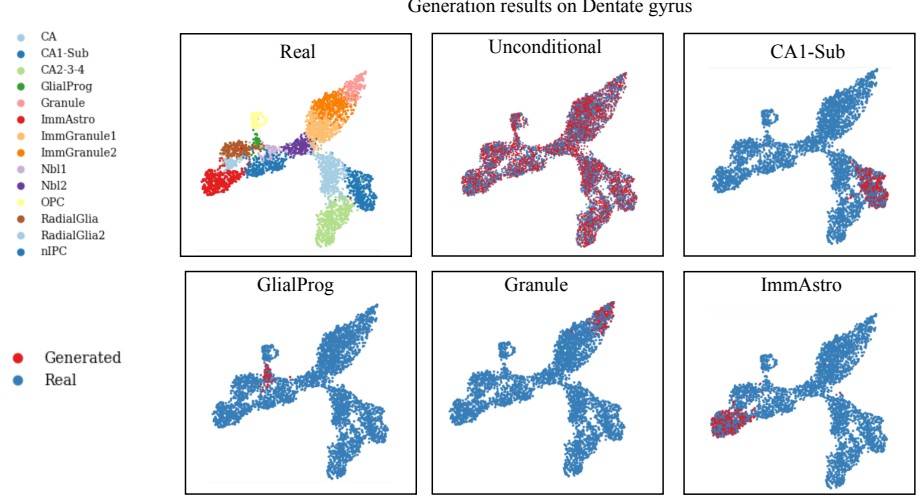

Figure A3: Uni-modal generation of scRNA-seq by CFGen on the Dentate gyrus dataset. Real and generated cells are embedded together and visualized as 2D UMAP coordinates.

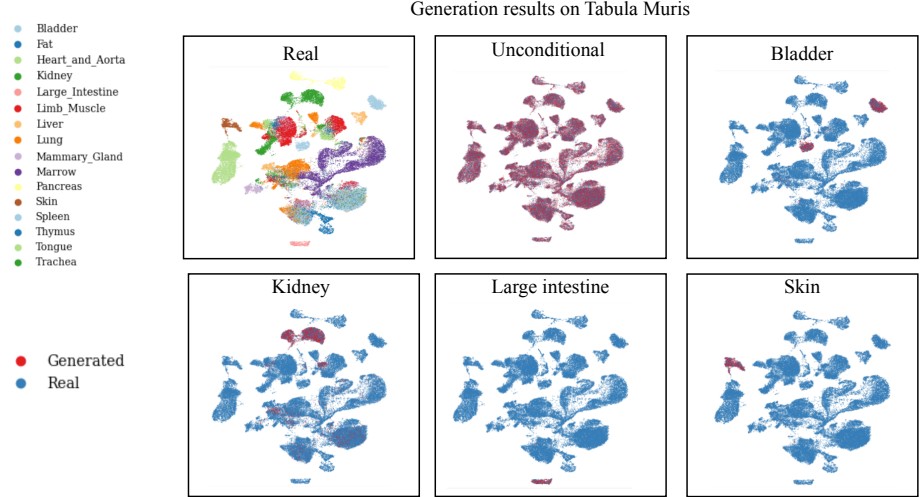

Figure A4: Uni-modal generation of scRNA-seq by CFGen on the Tabula Muris dataset. Real and generated cells are embedded together and visualized as 2D UMAP coordinates.

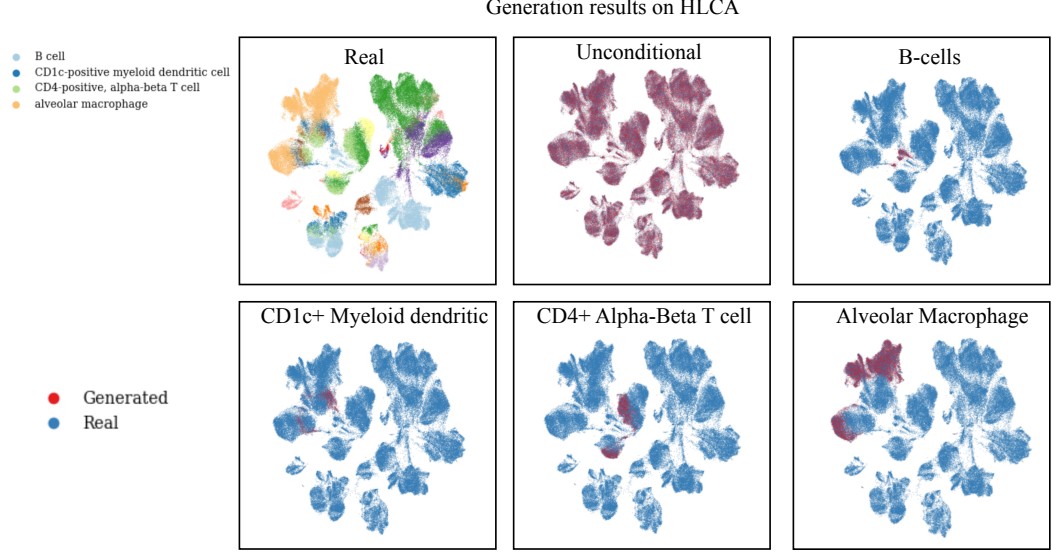

Figure A5: Uni-modal generation of scRNA-seq by CFGen on the HLCA dataset. Real and generated cells are embedded together and visualized as 2D UMAP coordinates.

### H.3  COMPARISON BETWEEN CFGEN, SCVI, AND SCDIFFUSION ON THE HLCA AND TABULA MURIS DATASETS

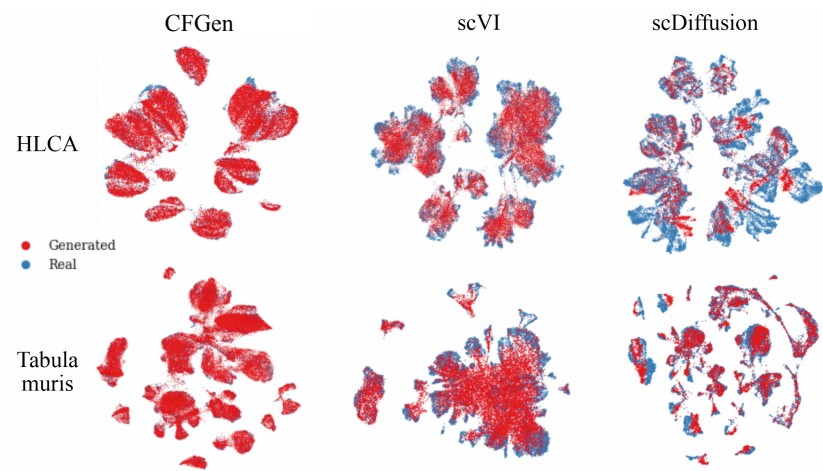

Figure A6: Qualitative comparison of the generation results of CFGen, scVI, and scDiffusion on the HLCA and Tabula Muris datasets. Comparison is performed by evaluating the similarity of the generated results to real cells. Real and generated cells for all models are embedded together and visualized as 2D UMAP coordinates.

### H.4  ADDITIONAL RESULTS ON MULTI-MODAL GENERATION

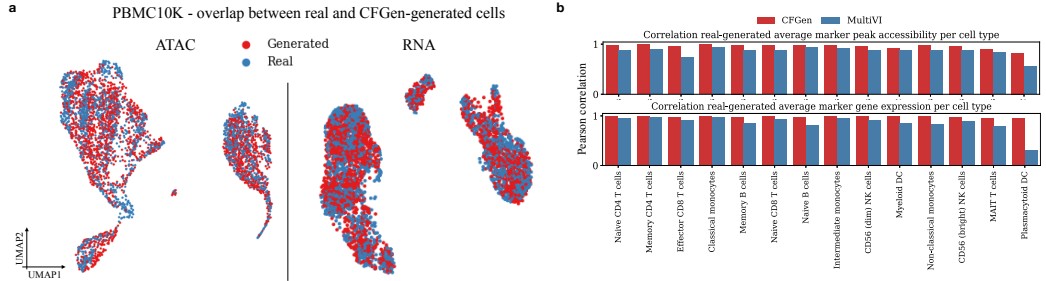

Figure A7: **(a)** 2D UMAP overlap between real and generated cells across modalities on the PBMC10K dataset. **(b)** Pearson correlation between average cell-type-specific marker peak accessibility and marker gene expression between real data and samples from CFGen and MultiVI.

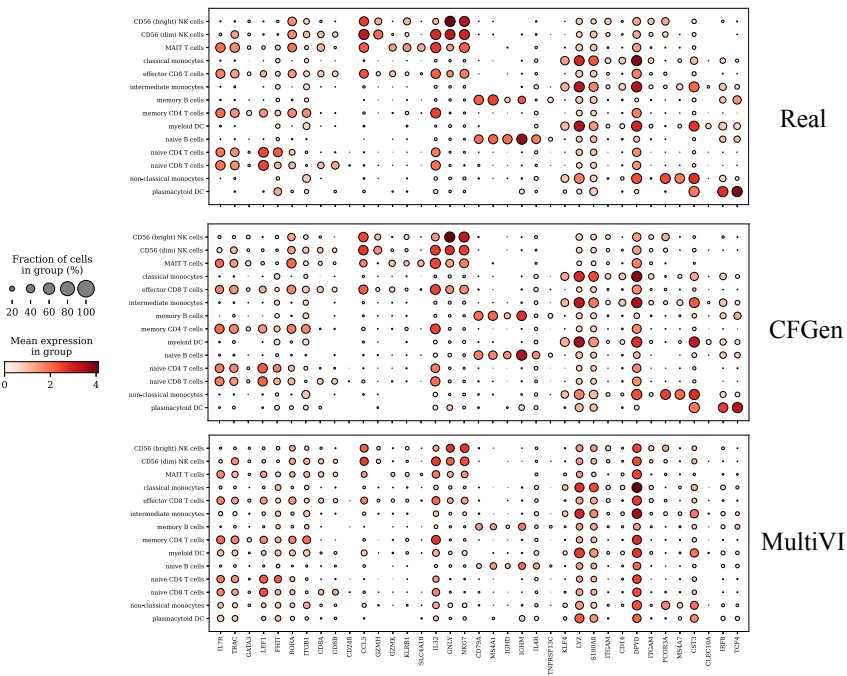

Figure A8: Average marker expression per cell type in real and generated data on the PBMC10k dataset. **x-axis** - marker genes. **y-axis** - cell types.

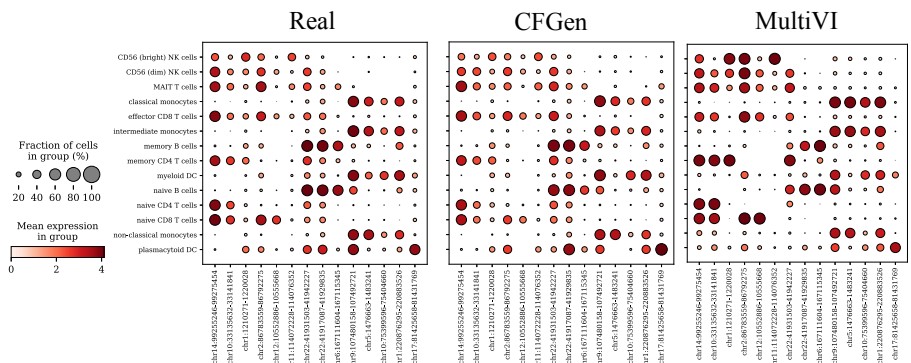

Figure A9: Average number of cells with accessible marker peaks per cell type in real and generated data on the PBMC10k dataset. **x-axis** - marker peaks. **y-axis** - cell types.

## H.5 ADDITIONAL RESULTS ON DATA AUGMENTATION

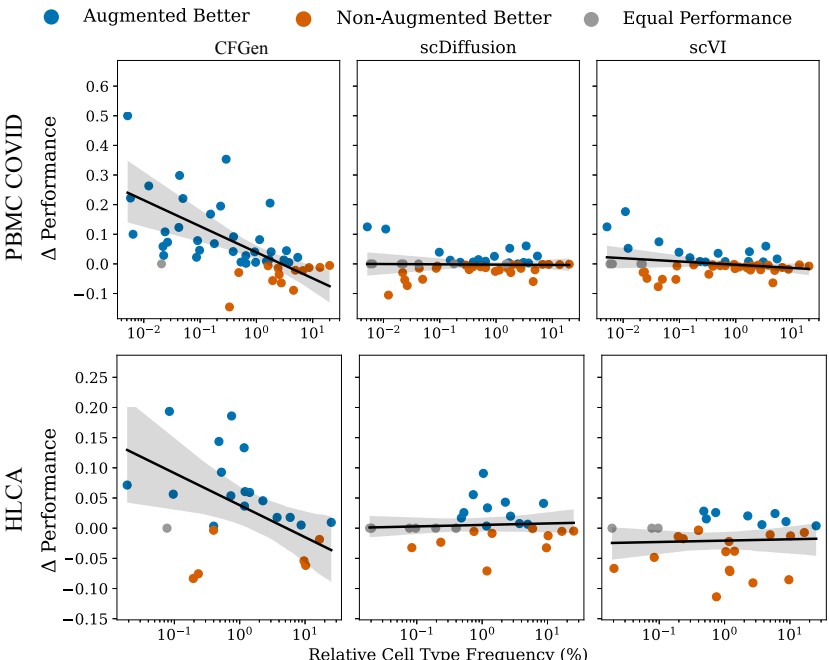

Figure A10: Extension of Fig. 4. Comparison of CFGen with scDiffusion and scVI on boosting the scGPT classifier performance in terms of recall on rare cell types.

Together with scGPT, in Fig. A11 we investigate if using CFGen to augment individual cell types improves the performance of a linear classifier like CellTypist (Cippà & Mueller, 2023). We obtain a similar result as scGPT, with the recall performance on real cell types improving after augmentation (hence a negative correlation between the performance improvement and the cell type frequency).

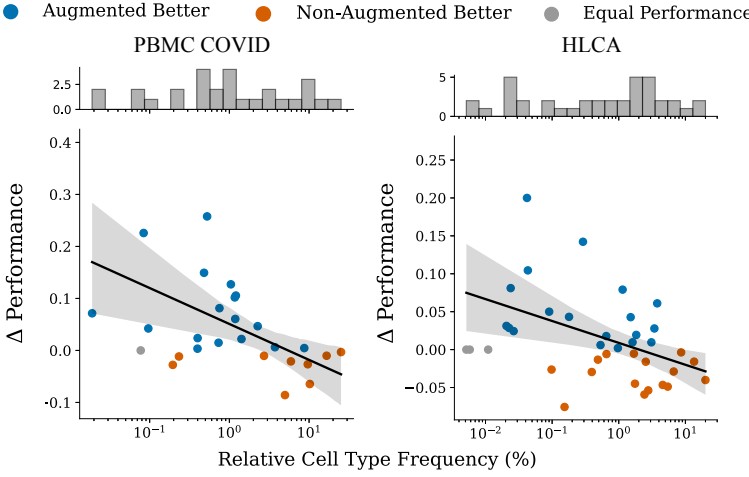

Figure A11: Performance improvement of a linear cell type classifier after data augmentation. The plot displays the cell-type classification performance difference in terms of recall as a function of cell type frequency before and after augmentation on PBMC COVID and HLCA datasets. As a classifier, we use CellTypist (Cippà & Mueller, 2023), which is based on logistic regression. The held-out set includes cells from 20% of donors for both datasets.

For a better appreciation of the classification improvement of single cell type categories by scGPT, we include Table 7 and Table 8.

Table 7: Table reporting the cell type classification recall of scGPT before (Recall Base) and after (Recall Aug) augmentation on the PBMC covid dataset. The Relative Frequency (%) column reports how rare a certain cell type is in the dataset. For each row, we highlight which setting leads to the highest recall.

| Cell Type | Relative Frequency (%) | Recall Base | Recall Aug |
|---|---|---|---|
| naive thymus-derived CD4-positive, alpha-beta ... | 25.18 | 0.87 | **0.90** |
| classical monocyte | 16.61 | 0.98 | 0.98 |
| natural killer cell | 10.22 | **0.91** | 0.90 |
| CD4-positive helper T cell | 9.63 | **0.80** | 0.75 |
| naive thymus-derived CD8-positive, alpha-beta ... | 8.70 | **0.83** | 0.79 |
| naive B cell | 5.92 | 0.96 | **0.98** |
| CD8-positive, alpha-beta cytotoxic T cell | 4.98 | **0.78** | 0.67 |
| non-classical monocyte | 3.73 | 0.95 | **0.96** |
| central memory CD8-positive, alpha-beta T cell | 2.72 | **0.45** | 0.33 |
| regulatory T cell | 2.27 | 0.28 | **0.32** |
| conventional dendritic cell | 1.42 | 0.79 | **0.84** |
| CD16-negative, CD56-bright natural killer cell ... | 1.21 | 0.66 | **0.70** |
| gamma-delta T cell | 1.19 | 0.49 | **0.63** |
| effector memory CD8-positive, alpha-beta T cell ... | 1.17 | 0.21 | **0.27** |
| class switched memory B cell | 1.04 | 0.51 | **0.71** |
| B cell | 0.75 | 0.21 | **0.39** |
| mucosal invariant T cell | 0.73 | 0.69 | **0.70** |
| CD4-positive, alpha-beta cytotoxic T cell | 0.53 | 0.06 | **0.13** |
| effector memory CD8-positive, alpha-beta T cell | 0.48 | 0.10 | **0.23** |
| plasmacytoid dendritic cell | 0.40 | 1.00 | 1.00 |
| platelet | 0.40 | 1.00 | 1.00 |
| plasma cell | 0.23 | **0.97** | 0.90 |
| hematopoietic precursor cell | 0.20 | **0.97** | 0.86 |
| mature NK T cell | 0.10 | 0.00 | **0.16** |
| innate lymphoid cell | 0.08 | 0.21 | **0.40** |
| erythrocyte | 0.08 | 1.00 | 1.00 |
| dendritic cell | 0.02 | 0.47 | **0.80** |
| plasmablast | 0.02 | 0.93 | **1.00** |
| granulocyte | 0.00 | 1.00 | 1.00 |

Table 8: Table reporting the cell type classification recall of scGPT before (Recall Base) and after (Recall Aug) augmentation on the HLCA dataset. The Relative Frequency (%) column reports how rare a certain cell type is in the dataset. For each row, we highlight which setting leads to the highest recall.

| Cell Type | Relative Frequency (%) | Recall Base | Recall Aug |
|---|---|---|---|
| alveolar macrophage | 20.00 | 0.95 | 0.95 |
| type II pneumocyte | 13.51 | 0.99 | 0.99 |
| respiratory basal cell | 8.63 | **0.92** | 0.90 |
| ciliated columnar cell of tracheobronchial tree | 6.67 | **0.97** | 0.94 |
| nasal mucosa goblet cell | 5.43 | 0.88 | **0.89** |
| CD8-positive, alpha-beta T cell | 4.93 | **0.89** | 0.87 |
| club cell | 4.57 | **0.62** | 0.53 |
| elicited macrophage | 3.77 | 0.70 | 0.70 |
| CD4-positive, alpha-beta T cell | 3.43 | 0.59 | **0.64** |
| vein endothelial cell | 3.09 | 0.93 | **0.94** |
| capillary endothelial cell | 2.77 | **0.92** | 0.85 |
| alveolar type 2 fibroblast cell | 2.54 | **0.94** | 0.90 |
| classical monocyte | 2.43 | 0.87 | 0.87 |
| CD1c-positive myeloid dendritic cell | 1.95 | **0.73** | 0.64 |
| pulmonary artery endothelial cell | 1.83 | 0.68 | **0.75** |
| lung macrophage | 1.75 | 0.36 | **0.57** |
| type I pneumocyte | 1.69 | 0.94 | **0.95** |
| non-classical monocyte | 1.61 | **0.55** | 0.54 |
| natural killer cell | 1.51 | 0.81 | **0.83** |
| multi-ciliated epithelial cell | 1.14 | 0.55 | **0.66** |
| endothelial cell of lymphatic vessel | 0.97 | 0.91 | **0.93** |
| epithelial cell of lower respiratory tract | 0.94 | 0.86 | **0.88** |
| mast cell | 0.65 | 0.96 | 0.96 |
| B cell | 0.65 | 0.88 | **0.90** |
| plasma cell | 0.53 | 0.98 | 0.98 |
| alveolar type 1 fibroblast cell | 0.49 | **0.82** | 0.79 |
| bronchus fibroblast of lung | 0.39 | 0.67 | **0.77** |
| respiratory hillock cell | 0.39 | 0.78 | **0.82** |
| tracheobronchial smooth muscle cell | 0.33 | **0.78** | 0.65 |
| epithelial cell of alveolus of lung | 0.29 | 0.18 | **0.55** |
| bronchial goblet cell | 0.23 | 0.04 | **0.11** |
| plasmacytoid dendritic cell | 0.18 | 0.87 | **0.94** |
| acinar cell | 0.15 | 0.67 | **0.81** |
| lung pericyte | 0.10 | 0.88 | **0.89** |
| ionocyte | 0.09 | 0.77 | **0.85** |
| T cell | 0.09 | **0.57** | 0.55 |
| tracheobronchial serous cell | 0.05 | 0.43 | **0.64** |
| myofibroblast cell | 0.04 | 0.25 | **0.69** |
| conventional dendritic cell | 0.04 | 0.58 | **0.81** |
| mucus secreting cell | 0.03 | 0.61 | 0.61 |
| dendritic cell | 0.02 | 0.46 | **0.70** |
| mesothelial cell | 0.02 | 0.91 | **1.00** |
| smooth muscle cell | 0.02 | 0.09 | **0.15** |
| lung neuroendocrine cell | 0.02 | 0.97 | 0.97 |
| brush cell of tracheobronchial tree | 0.01 | 0.21 | **0.37** |
| stromal cell | 0.01 | 0.35 | **0.88** |
| fibroblast | 0.01 | 0.30 | **0.60** |
| hematopoietic stem cell | 0.01 | 0.78 | **0.89** |
| tracheobronchial goblet cell | 0.01 | 0.00 | **0.50** |

## H.6 MISSING GENE IMPUTATION WITH CFGEN

In the scVI paper (Lopez et al., 2018), 10% of data entries are masked and set to zero, with the model trained on this corrupted data. During inference, masked cells are passed through the encoder, and latent codes $\mathbf{z} \sim q_\psi(\cdot|\mathbf{x})$ are sampled from the posterior. The VAE, trained to handle noisy inputs, decodes $\mathbf{z}$ to infer masked counts. Similarly, we propose an imputation strategy using CFGen as follows:

- Train CFGen on noisy data.
- Encode a noisy input $\mathbf{x}$ into the latent representation $\mathbf{z}_1 = f_\psi(\mathbf{x})$.
- Invert the generative flow to compute $\mathbf{z}_0 = \phi_0(\mathbf{z}_1)$, mapping $\mathbf{z}_1$ to noise.
- Sample around $\mathbf{z}_0$ as $\mathbf{z}_0' \sim \mathcal{N}(\mathbf{z}_0, \sigma^2 \mathbf{I})$.
- Transport $\mathbf{z}_0'$ back to $\mathbf{z}_1' = \phi_1(\mathbf{z}_0')$, then decode to impute gene values for $\mathbf{x}$.

We tested this strategy on four datasets, masking 10% of the counts. Fig. A12 shows that our predictions correlate with pre-masking data, and Table 9 demonstrates superior imputation accuracy compared to scVI in three out of four datasets (Pearson correlation, mean absolute distance).

Table 9: Mean distance and correlation between real and imputed genes by scVI and CFGen.

| | Mean $L_1$ distance real-imputed counts ($\downarrow$) | | | | Pearson correlation real-imputed counts ($\uparrow$) | | | |
| | PBMC3K | Dentate gyrus | HLCA | T. Muris | PBMC3K | Dentate gyrus | HLCA | T. Muris |
|---|---|---|---|---|---|---|---|---|
| CFGen | **1.21** | 0.42 | **3.21** | **4.81** | **0.68** | 0.56 | **0.83** | **0.86** |
| scVI | 1.47 | **0.35** | 4.43 | 6.08 | 0.61 | **0.58** | 0.75 | 0.79 |

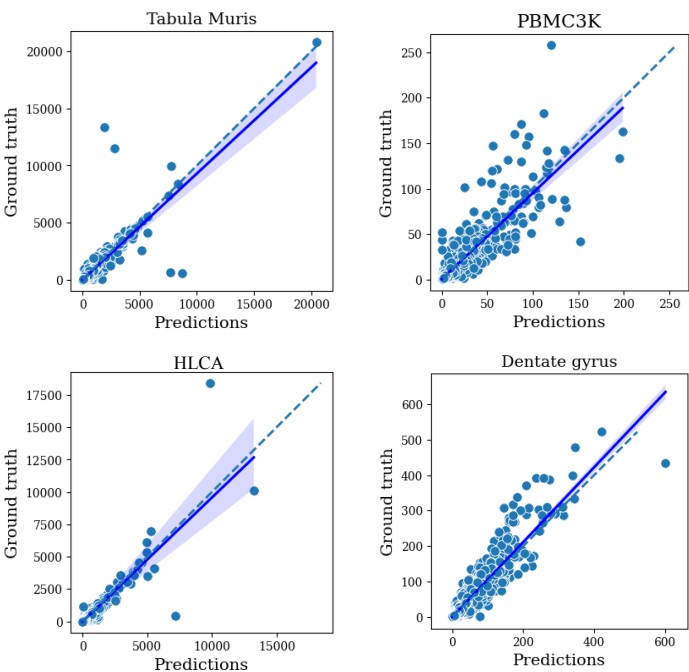

Figure A12: Scatterplot between imputed and real gene expression values before masking across datasets. Correlations can be found in Table 9.

In Fig. A13, we study how the quality of the imputation by CFGen varies as a function of the amount of noise used to sample around $\mathbf{z}_0$. Notably, a higher noise level leads to worse imputation results, since the generative modeling aspect takes over and samples a completely new cell, which loses the structure of the originally encoded noisy observation. Specifically, Fig. A13 highlights that $\sigma$ should

remain below 0.1 to avoid sampling distant $\mathbf{z}_0'$ values, which generate unrelated cells and disrupt correlations with original gene expressions.

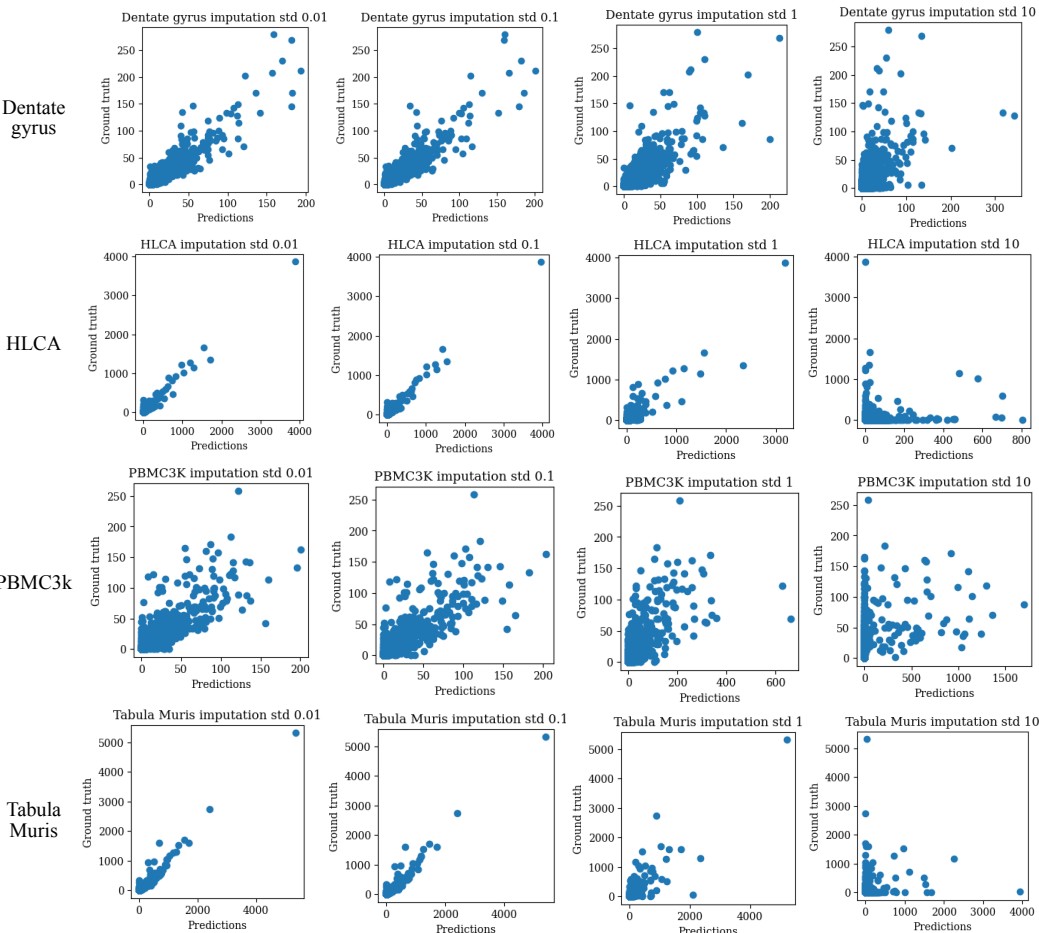

Figure A13: Correlation between the CFGen-imputed and real gene expression before masking as a function of the variance of the added noise. Rows represent different datasets, columns represent the standard deviation of the noise used to sample around the latent representation of the cell. Perfect correlation along the bisector is the best possible imputation result.

## H.7 ADDITIONAL RESULTS MULTI-LABEL GENERATION

Table 10: Extension to Fig. 3. We train a 3-layer MLP with a softmax head on the real data to predict the classes of the two attributes considered for each dataset. For different levels of the combination of guidance weights, the classifier is applied to predict the average probability that the generated observations are of a certain guidance class. When guided on a single attribute, it is expected that the generated cells are assigned with high probability only to the class of such an attribute. As guidance strength increases for the counterpart attribute, CFGen models the intersections between attributes increasingly better and, therefore, enables high classification probability for both guiding labels.

| | NeurIPS | | | Tabula Muris | |
|---|---|---|---|---|---|
| Weights | $p(\text{CD14} + \text{M.})$ | $p(\text{donor 1})$ | Weights | $p(\text{Tongue})$ | $p(\text{18-M-52})$ |
| $\omega_{\text{donor}} = 0$ $\omega_{\text{cell type}} = 1$ | 0.98 | 0.40 | $\omega_{\text{mouse ID}} = 0.0$ $\omega_{\text{tissue}} = 1$ | 0.98 | 0.19 |
| $\omega_{\text{donor}} = 1$ $\omega_{\text{cell type}} = 1$ | 0.96 | 0.87 | $\omega_{\text{mouse ID}} = 1$ $\omega_{\text{tissue}} = 1$ | 0.98 | 0.69 |
| $\omega_{\text{donor}} = 5$ $\omega_{\text{cell type}} = 1$ | 0.96 | 1.00 | $\omega_{\text{mouse ID}} = 2.5$ $\omega_{\text{tissue}} = 1$ | 0.96 | 0.97 |

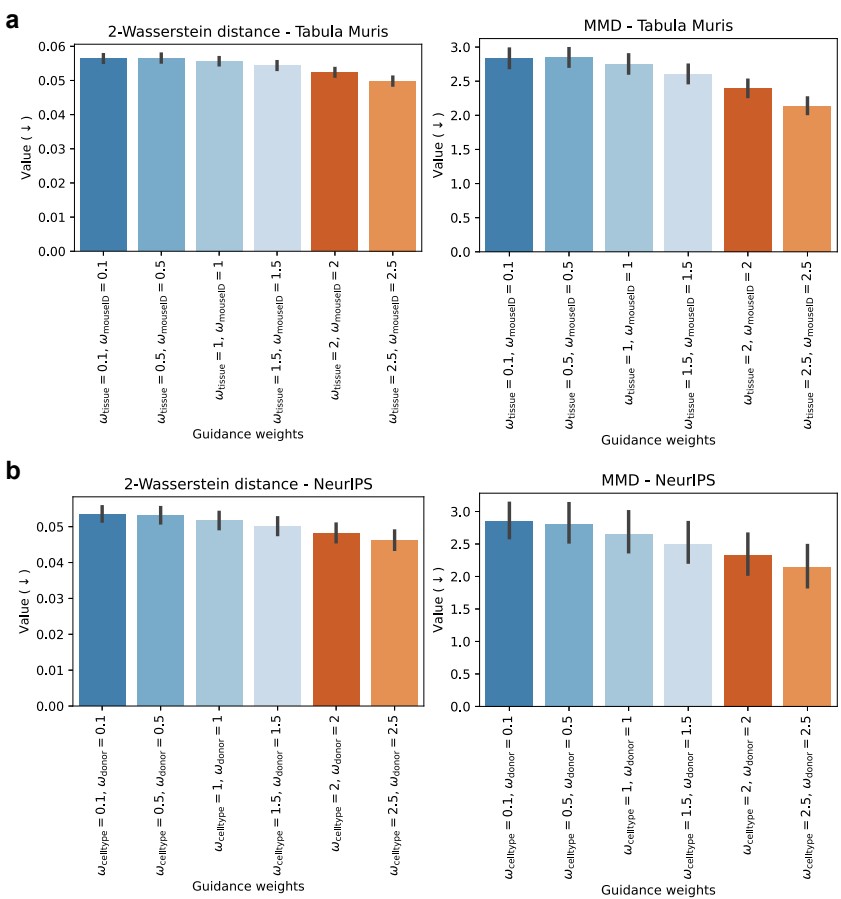

Figure A14: Performance on the generation of the intersection of attributes on the Tabula Muris (top) and NeurIPS (bottom) datasets based on distributional metrics. On the x-axis, we increase the guidance parameters for both conditioning attributes.

## H.8 ADDITIONAL RESULTS ON MULTI-ATTRIBUTE GUIDANCE RESULTS

In batch correction, cells are transported to noise and then back to data, guided by both a biological and a target batch covariate. The guidance strength parameters $\omega_{bio}$ and $\omega_{batch}$ control the emphasis on biological conservation and batch correction, respectively.

Table 11: The average batch correction and bio conservation metrics from the scIB package evaluated at different levels of guidance strength.

| Guidance weights | C. Elegans | | NeurIPS | |
|---|---|---|---|---|
| | Batch Correction | Bio Conservation | Batch Correction | Bio Conservation |
| $\omega_{bio}=0\,\omega_{batch}=0$ | 0.48 | 0.55 | 0.32 | 0.63 |
| $\omega_{bio}=1\,\omega_{batch}=1$ | 0.67 | 0.55 | 0.61 | 0.64 |
| $\omega_{bio}=1\,\omega_{batch}=2$ | 0.68 | 0.54 | 0.63 | 0.61 |
| $\omega_{bio}=2\,\omega_{batch}=1$ | 0.68 | 0.63 | 0.64 | 0.73 |
| $\omega_{bio}=2\,\omega_{batch}=2$ | 0.69 | 0.63 | 0.64 | 0.71 |
| $\omega_{bio}=2\,\omega_{batch}=3$ | 0.69 | 0.64 | 0.65 | 0.70 |
| $\omega_{bio}=3\,\omega_{batch}=2$ | 0.70 | 0.67 | 0.65 | 0.77 |
| $\omega_{bio}=3\,\omega_{batch}=3$ | 0.70 | 0.68 | 0.66 | 0.75 |
| $\omega_{bio}=3\,\omega_{batch}=4$ | 0.70 | 0.67 | 0.66 | 0.73 |
| $\omega_{bio}=4\,\omega_{batch}=4$ | 0.70 | 0.69 | 0.67 | 0.77 |

If one relies on the scIB metric in Table 11, computed for different guidance strength parameters, the highest possible guidance strengths may appear optimal, as they yield the best aggregation within cell types and batches. However, Fig. A15 and Fig. A16 demonstrate that scIB metrics can be misleading and should be complemented by qualitative evaluation. Excessive guidance in the translation task causes an unnatural collapse of variability beyond what is explained by batch and biological annotations.

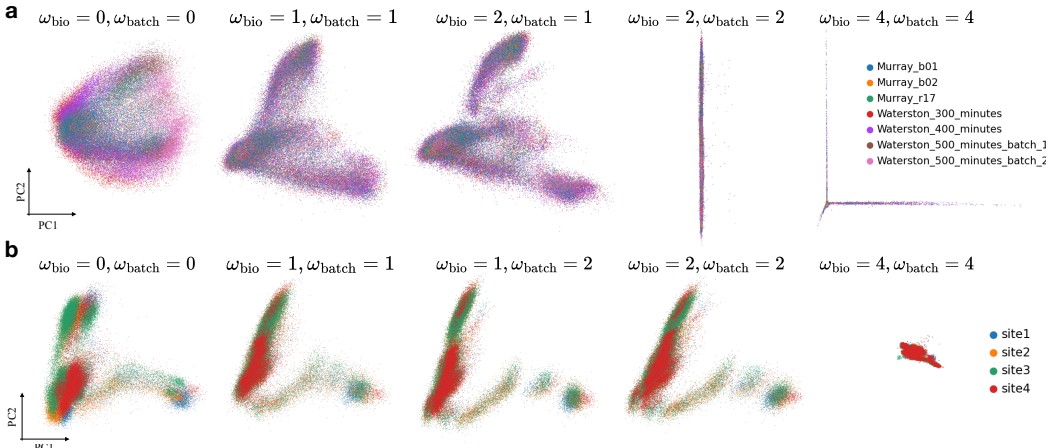

Figure A15: The PCA plot of generated cells colored by batch for the C.Elegans (**a**) and NeurIPS (**b**) datasets. Each column represents a different combination of guidance strength values.

We found that guidance strength parameters in the range $\omega_{bio}, \omega_{batch} \in \{1, 2\}$ effectively preserve biological signal while performing batch correction without over-squashing cell representations. An example of these unwanted effects is illustrated in Fig. A16, where excessive biological preservation results in unnatural clustering for both datasets.

Moreover, the severity of batch effects in the data should guide parameter selection. In the C.Elegans dataset, where batch effects are mild, we select $\omega_{bio} = 2, \omega_{batch} = 1$ as they provide better performance than $\omega_{bio} = 1, \omega_{batch} = 2$ (Table 11). Conversely, in the NeurIPS dataset, we observe the opposite effect and thus select $\omega_{bio} = 1, \omega_{batch} = 2$. As shown in Fig. A16, choosing $\omega_{bio} > 1$ leads to an unnatural biological structure, violating smooth temporal single-cell trajectories.

In conclusion, we recommend first assessing the severity of batch effects in the dataset, and then systematically sweeping over combinations of guidance weights. The optimal configuration should maximize scIB metric values while maintaining realistic single-cell representations.

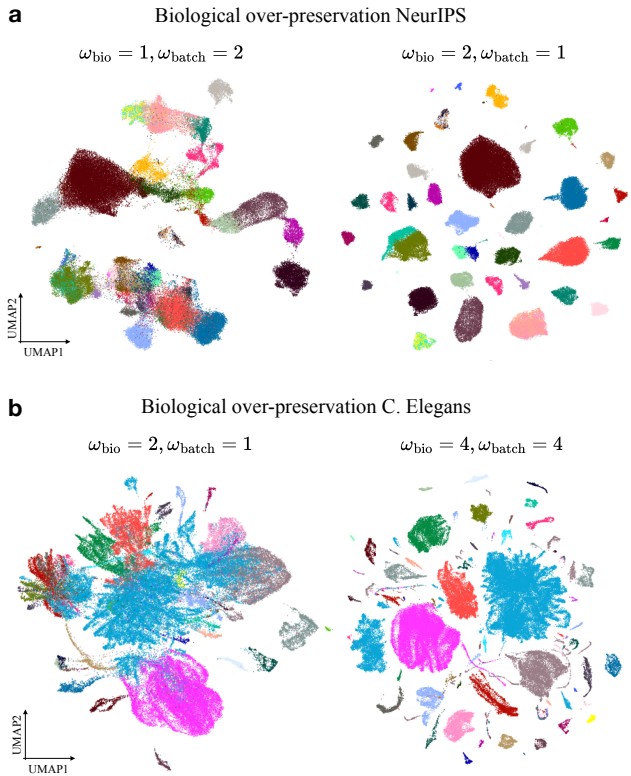

Figure A16: The UMAP plot of generated cells colored by batch for the (**a**) NeurIPS and (**b**) C.Elegans datasets. We show one example of generation with a reasonable guidance scheme (left columns) and one with a guidance scheme causing unrealistic cell type distributions (right).

### H.9    LIBRARY SIZE INFLUENCE IN SINGLE-CELL ATLASES

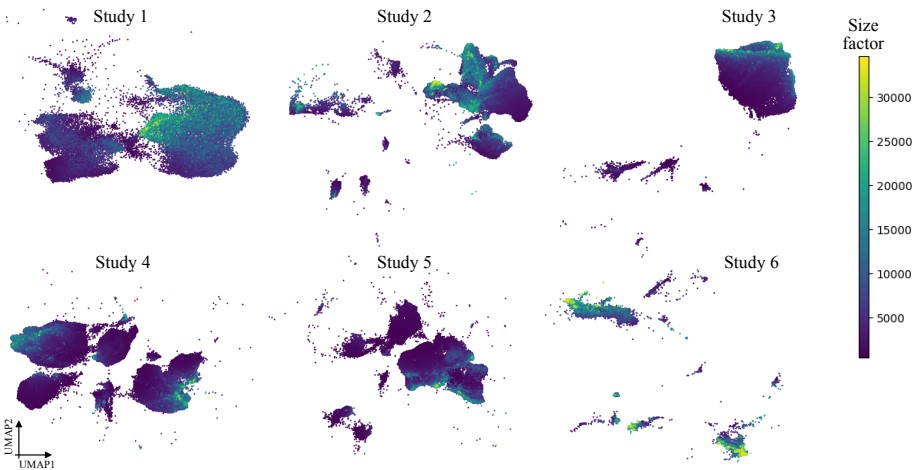

Figure A17: UMAP plots of six studies included in the HLCA dataset colored by size factor.

