# OpenReview forum: "Multi-Modal and Multi-Attribute Generation of Single Cells with CFGen"
_ICLR.cc/2025/Conference — ICLR 2025 Poster_

### Official Review · Reviewer_FqvM · 2024-11-03

**Soundness:** 2
**Presentation:** 3
**Contribution:** 2
**Rating:** 6
**Confidence:** 4

**Summary:**

In summary, I initially rate this paper as  "5: marginally below the acceptance threshold". But I'm open to increase my score if authors properly answer my doubts in the rebuttal.

Summary of paper: The paper proposes a generative model for scRNA as well as accessibility modalities. The model can take in a combination of attributes, which suits the biological settings where for each cell only a subset of attributes are available. The method is evaluated in generation, handling label imbalance in cell type classification for rate cell types, and batch correction.

**Strengths:**

- The model is tailored to real biological settings: it handles 2 modalities (scRNA and ATAC) and any number of attributes.
- The results properly support the good performance of the method.
- Besides generation power, two very interesting applications are demonstrated: handing rare cell types in cell type classification and batch correction.

**Weaknesses:**

- Handling discrete count data via negative binomial distribution is presented as a "contribution" of this paper. But there is a plethora of methods that make use of negative binomial (or alternatives like poisson distribution) to handle count data as well as over-dispersion. So why should it be listed as a contribution of this paper?
- According to the paper, "... the proposed factorisation is novel". In the factorisation of Eq. 5 what is the rational behind conditioning the latent factor z on library size?
- In proposition 1, the attributes $y_1$, $y_2$, ... are assumed to be conditionally independent given $z$, but with the factorisation of Eq. 5 the attributes are connected to $z$, hence $z$ forms a V-structure which according to d-separation causes the attributes to be dependant given $z$ ?
- Regarding the proposed guidance scheme, the only difference to the normal classifier-free guidance is that only some attributes (and one attribute during training) is fed to the decoder. Is this approach equivalent to the normal classifier-free guidance with all attributes plus some attributes being randomly dropped out? Even if so, it wouldn't decrease the value of the proposed method.
- In Table 1 scDiffusion is heavily outperformed by the proposed method, but one may say diffusion models may perform on par with flow matching (apart from training stability etc.). In the paper I'd recommend providing an explanation for the superior performance of the proposed method compared to scDiffusion.

**Questions:**

Please see the "Weaknesses" part.

---

> ### Author Response · Authors · 2024-11-21
> **Rebuttal 1**
>
> We thank Reviewer FqvM for investing the time in reviewing our work. We are glad to read positive comments about our presentation and contribution. We also appreciate the constructive criticism and feedback we received. We hope our clarifications and changes to the manuscript will contribute to a more positive assessment of our manuscript.
>
> > Handling discrete count data via negative binomial distribution is presented as a "contribution" of this paper. Why?
>
> We apologize if our phrasing generated some lack of clarity. We will elaborate more on our contribution here. Our goal is not to claim that we are the first to introduce the concept of modeling scRNA-seq and ATAC-seq data with discrete distributions, this is standard practice in many settings. However, most existing models leveraging such a likelihood formulation are VAE-based approaches (like scVI) with a strong focus on learning biologically meaningful and batch-free representations rather than optimizing for synthetic data generation. In other words, scVI and its relatives do not aim at fully regularizing the latent space to standard Gaussian and allow the retaining of biological structure for interpretable representations. This, however, hinders the generation potential.
>
> On the other side of the spectrum, there are models like scGAN and scDiffusion which, unlike scVI, optimize for pure generation. Such models do not factor the probabilistic properties of the data into their models and rely on powerful generative schemes to learn a continuous approximation of the data. The lack of consideration of such properties is a disadvantage when modeling complex domains like discrete biological data since they exhibit skewness, sparsity and overdispersion that do not resonate well with continuous models.
>
> Our CFGen contributes to implementing the *best of both worlds*. The model is founded on rigorous considerations of the data characteristics (cells are modeled using a negative binomial likelihood) while optimizing for synthetic data generation (unlike common scVAE models) and still providing representation-learning-based tasks available in scVI, like batch correction. In summary, our contribution is not the choice of the likelihood itself, but rather its combination with a versatile flow-based generative framework encompassing both generation and manipulation of single-cell representations.
>
> > According to the paper, "... the proposed factorisation is novel". In the factorisation of Eq. 5, what is the rational behind conditioning the latent factor z on library size?
>
> To answer this question, we start comparing how we handle the size factor to the scheme implemented by models like scVI. Single-cell VAEs provide an option to learn a log-normal distribution over the size factor that one can sample from. This is achieved similarly as we propose, namely by fitting a log-normal distribution over the library size in the data (it can also be conditional on the batch variable). The main difference between our approach and scVI is that in the latter the latent cellular state variable and the size factor are sampled *independently*, in CFGen we provide the option to bias the sampling by the size factor. In other words, we sample a latent state $\mathbf{z}$ that accounts for cell size, while scVI does not.
>
> Why is this important? To generate from scVI, one would sample a batch, sample a size factor conditioned on the batch, sample a latent code independently and then generate a cell using the conditional decoder's output scaled by the sampled size factor (which is not involved in the decoding, only in the scaling). In CFGen, we would instead bias the sampling of the latent code on the library size. Now, if the size factor is somewhat uniform within the batch, the approach in scVI might work. However, if the batch key represents a coarse annotation (e.g., the source study of a dataset in modern atlases), the library size can vary significantly within the batch. This variability reflects differences inherent to the batch annotation (see Fig. A18 for examples using the Human Cell Atlas dataset). Sampling a cell state and library size independently in such cases could result in scaling a decoded cell state by an incompatible size factor, producing unrealistic outcomes. Instead, conditioning on the size factor is more appropriate, as it biases the sampling of the latent state toward regions exhibiting a specific size factor. This approach can be combined with coarsely annotated variables for a more targeted conditional generation.
>
> In summary, conditioning on size factor steers the generated state towards regions of the cell space where scaling for such size factor makes sense.

---

> ### Author Response · Authors · 2024-11-21
> **Rebuttal 2**
>
> > Conditional independence assumption.
>
> Thank you for the question. First, we would like to point out that the definition in Eq. 5 is the joint distribution for the generative model conditioned on a single attribute. The (conditional) independence assumption is a standard practice in multi-attribute generative modeling [1, 2, 3, 4], where the central mathematical formulation relies on rewriting the multi-attribute conditional probability distribution as a factorization over single-attribute conditional densities:
>
> $p(\mathbf{z}|y_1, y_2,...,y_K) \propto p(\mathbf{z})\prod_{i=1}^{K}\frac{p(\mathbf{z}|y_i)}{p(\mathbf{z})}$.
>
> To obtain the product, one must first assume conditional independence of the factors $y_i$ between each other given an observed $\mathbf{z}$. In the context of score-based models, this formulation allows to expression of the conditional score as a composition of an unconditional model and single-attribute conditional models (see Eq. 17 in the manuscript). In these regards, our assumption follows the line of established works in the field of generative models, where we extend the idea to Flow Matching to fit in our modeling framework (see Prop. 5 and the Appendix sections A.2 to A.4 for a detailed breakdown of the components).
>
> To add more intuition to the above formulation, we assume that, given a state $\mathbf{z}$ and two conditioning variables $y_1$ and $y_2$, observing $\mathbf{z}$ sufficiently captures the interactions between $y_1$ and $y_2$. None of the cited works have raised concerns regarding the arising of V-structure dependencies violating the assumption. In our setting, we can intuitively think that, provided that we have a rich cell representation $\mathbf{z}$, if we can observe $\mathbf{z}$, then adding $y_1$ does not give more information about $y_2$, and vice versa. Hence, $p(y_1|\mathbf{z}, y_2)=p(y_1|\mathbf{z})$. For example, if $y_1$ is cell type and $y_2$ is batch, then we are assuming that if one can observe a cell state $\mathbf{z}$ and the batch $y_2$, the information from the variable $y_2$ does not give any additional information on $y_1$ that is not already contained in $\mathbf{z}$. Since we model a very informative $\mathbf{z}$, our results as well as the cited papers show evidence that the assumption is reasonable.
>
>
> > Guidance scheme.
>
> Our method is an extension to classifier-free guidance using multiple attributes. From a theoretical perspective, the model differs from the standard classifier-free guidance approach in that it models the contribution of multiple single-attribute flows additively on the final generative model. More in detail, in Section A.4 we proved that learning and simulating the additive vector field in Eq. 8 allows us to generate observations compositionally on the chosen attributes.
>
> Compositionality means that one can use a single model to generate observations from either all conditions, subsets thereof and unconditionally. From a training perspective, the difference from using all attributes at a time as in classifier-free guidance is indeed that each step only sees one attribute. But the differences also involve sampling, where we compose $K$ different vector fields weighted by their respective $\omega_i$. Unlike standard classifier-free guidance or simple joint conditioning by multiple variables, here we are able to tune up or down the effect of individual attributes without altering the others, which is useful in single-cell tasks like batch effect removal, where the extent of correction can be modulated by how strong the batch component is in the dataset.

---

> ### Author Response · Authors · 2024-11-21
> **Rebuttal 3**
>
> > Reasons for the performance of scDiffusion
>
> Thank you for your suggestion. As we also described in our first answer, diffusion models such as scDiffusion indeed optimize for the generation task. However, in the context of scRNA-seq, some modeling choices may be responsible for hindering the performance of the generation of scRNA-seq data:
> * scDiffusion does not take into account important properties of single-cell data, such as sparsity, overdispersion and discreteness. Although normalizing the data is an option to ensure continuity, most normalization methods preserve zeros in the data and a non-linear mean-variance trend. Continuous models usually benefit from centered and non-sparse input, making the structure of scDiffusion with a continuous decoder sub-optimal.
> * The model relies on classifier-based guidance, therefore conditional sampling is heavily dependent on the performance of the classifier on individual labels. This structurally challenges the application of scDiffusion to rare cell type generation or any guiding schemes using hard-to-classify attributes.
> * For very smaller datasets like PBMC3k, training SDE-based diffusion models is empirically complex and unstable.
>
> In our framework, all these aspects are overcome by training a latent Flow Matching model with a discrete likelihood scheme and classifier-free guidance. We add a description of potential limitations in the scDiffusion model to Appendix D.5.
>
> [1] Liu, Nan, et al. "Unsupervised compositional concepts discovery with text-to-image generative models." Proceedings of the IEEE/CVF International Conference on Computer Vision. 2023.
>
> [2] Shi, Changhao, et al. "Exploring compositional visual generation with latent classifier guidance." Proceedings of the IEEE/CVF Conference on Computer Vision and Pattern Recognition. 2023.
>
> [3] Du, Yilun, Shuang Li, and Igor Mordatch. "Compositional visual generation with energy based models." Advances in Neural Information Processing Systems 33 (2020): 6637-6647.
>
> [4] Liu, Nan, et al. "Compositional visual generation with composable diffusion models." European Conference on Computer Vision. Cham: Springer Nature Switzerland, 2022.

---

> > ### Author Response · Authors · 2024-11-24
> >
> > Dear Reviewer,
> >
> > Thank you once again for the time and effort invested in reviewing our paper. The insightful feedback provided has greatly contributed to strengthening the presentation of our work, particularly by enabling a more robust justification of our results and empirical observations.
> >
> > As we approach the final phase of the discussion period, we would like to confirm whether our additional elaborations have satisfactorily addressed the relevant doubts expressed by the reviewer regarding our model. If the rebuttal and responses have resolved all remaining concerns, we would be grateful if the reviewer might consider revising their score as suggested during the review stage. Naturally, we remain fully available to address any further questions or issues if necessary.
> >
> > We sincerely appreciate the reviewer's valuable input and guidance once more.
> >
> > Best regards,
> >
> > The Authors

---

> ### Comment · Reviewer_FqvM · 2024-11-25
>
> The authors addressed all of my major concerns. So I'm increasing my score from 5 to 6.

---

> > ### Author Response · Authors · 2024-11-25
> > **Thank you for your answer.**
> >
> > Dear Reviewer FqvM,
> >
> > We are very pleased to hear that our responses addressed your major concerns and resulted in a score increase. Thank you once again for your valuable feedback and insightful questions.
> >
> > Best regards,
> >
> > The Authors

---

### Official Review · Reviewer_eqfJ · 2024-11-04

**Soundness:** 3
**Presentation:** 3
**Contribution:** 3
**Rating:** 5
**Confidence:** 3

**Summary:**

The authors of this paper present CFGen, a flow-based generative model designed for multi-modal single-cell data. CFGen addresses the challenges of generating discrete, multi-modal data while allowing conditional generation based on various biological attributes. The model extends the flow matching framework to handle compositional guidance across multiple attributes and provides promising results on tasks like data augmentation, rare cell type classification, and batch correction.

**Strengths:**

- The authors nicely demonstrate practical applications of their method such as data augmentation in rare cell types, improving downstream classification, and performing batch correction.

- The idea to extend flow matching for generation with multiple attributes is interesting and important for single-cell data.

- The paper is well-written, the related work is appropriately referenced, and the experimental setup is detailed.

**Weaknesses:**

-  The authors do not discuss the computational complexity of the proposed method. A more detailed breakdown of computational requirements, including training and sampling times for the proposed method and the baselines, would improve the paper.

- One important task in single-cell data analysis is gene expression imputation, where missing or zero-inflated gene expression values are inferred to provide a more complete view of cellular states. It is unclear from the paper whether CFGen can effectively handle this task, given its focus on generating new cells rather than imputing missing data within existing cells. Could the authors clarify if CFGen’s architecture or the flow matching framework could be adapted for imputation?

**Questions:**

- Can CFGen be applied to gene expression imputation tasks? If so, could the authors describe how the current framework could handle imputation, or if modifications would be needed?

- Could the authors provide details about the computational complexity of the model?

---

> ### Author Response · Authors · 2024-11-21
> **Rebuttal 1**
>
> We acknowledge the Reviewer's feedback and are thankful for the positive comments on the quality of the work. We particularly value the received criticism, as it pushed us to extend the scope of our contribution to the gene imputation task. We hope Reviewer eqfJ will find our new results insightful.
>
> > The authors do not discuss the computational complexity of the proposed method. A more detailed breakdown of computational requirements, including training and sampling times for the proposed method and the baselines, would improve the paper.
>
> We included a new section to the Appendix (Section H.1) reporting a detailed breakdown of the runtime and computational complexity of our model. More specifically, we added information on CFGen's runtime in Fig. A1, Tab. 5 and Tab. 6. We describe the insights derived from our new results here:
> * **Fig A1** illustrates how the generation runtime changes as a function of hyperparameters. Since CFGen is a latent generative model, we consider different runtime curves for different latent space sizes as a function of the following hyperparameters. The size of the latent space significantly influences the runtime, since it establishes the size of the state space where the generative ODE is simulated. The sampling runtime increases approximately linearly with respect to the number of cells and genes, while neural-network-related hyperparameters appear not to impact the simulation time significantly.
> * In **Tab. 5** We compare training times between the different models across benchmarked datasets. Of course, smaller VAE-based models like scVI are faster to train than diffusion and Flow-Matching-based counterparts. Between CFGen and scDiffusion, the latter exhibits longer combined training times on larger datasets like Tabula Muris and HLCA, but also on PBMC10K. Importantly, scDiffusion's training requires optimizing an autoencoder, a denoising diffusion network and a cell type classifier used for guidance separately, while CFGen only trains an autoencoder and a Flow Matching model.
> * **Tab. 6** reports the sampling times of the trained generative models. Intuitively, VAE models are still faster, since they do not have to simulate generative ODEs and SDEs across time. However, our results also suggest that such VAEs are qualitatively and quantitatively limited on large datasets (Fig. A7) and downstream applications such as augmentation (Fig. A11) and multi-modal marker generation (Fig. A8-A9). CFGen can generate atlas-level datasets of >500k cells in around 8 seconds (see the HLCA column). Conversely, scDiffusion requires approximately half an hour for the task and is therefore not a sensible candidate for augmenting datasets with millions of cells. We highlight the aspects that make CFGen faster and more performing than scDiffusion in Appendix D.5.

---

> > ### Author Response · Authors · 2024-11-21
> > **Rebuttal 2**
> >
> > > It is unclear from the paper whether CFGen can effectively handle the imputation task, given its focus on generating new cells rather than imputing missing data within existing cells. Could the authors clarify if CFGen’s architecture or the Flow Matching framework could be adapted for imputation?
> >
> > We found that CFGen can be used for the imputation task. As a reminder, imputation in the scVI paper is performed by masking 10% of the data entries with zeros. Subsequently, the model is trained on the corrupted representation. During inference, masked cells are passed through the encoder and a latent code is sampled from the Gaussian posterior as $\mathbf{z} \sim q_\psi(\cdot|\mathbf{x})$, where $\mathbf{z}$ is the latent code and $\mathbf{x}$ is the data point. In other words, sampling around the mean of the posterior yields a representation that is mapped to a sensible imputed cell. In our settings, we do not have a posterior $q_\psi$, but we can still sample around an observation using our invertible flow. We propose the following workflow:
> > * Train CFGen on noisy data.
> > * Take a noisy input $\mathbf{\mathbf{x}}$ and encode it into a latent representation using the CFGen encoder $\mathbf{z_1} = f_{\psi}(\mathbf{x})$ (the subscript 1 is introduced for notational simplicity in later steps).
> > * Simulate the inverted flow $\mathbf{z_0}=\phi_0(\mathbf{z}_1)$ from timepoint 1 to 0. Simply put, we invert the generative flow, mapping $\mathbf{z}_1$ (deterministically) to its representation under the standard normal prior, similar to what we do in batch correction.
> > * We sample around $\mathbf{z}_0$ as $\mathbf{z}_0' \sim \mathcal{N}(\mathbf{z}_0, \sigma^2 I_d)$, where $I_d$ is the d-dimensional identity matrix and $\sigma^2$ is a pre-defined variance hyperparameter.
> > * The resulting representation is transported back simulating the flow to obtain $\mathbf{z}_1'=\phi_1(\mathbf{z}_0')$. Finally, we decode the extracted representation and obtained imputed genes.
> >
> > We applied our strategy to the four datasets benchmarked in the paper, which were first preprocessed by masking 10% of the counts randomly. In Fig. A13, we show that our model yields predictions correlated with the data before imputation, while Tab. 11 illustrates that CFGen imputes masked genes better than scVI in three datasets out of four in terms of Pearson correlation and mean absolute distance.
> >
> > Finally, in Fig. A14 we show that the value of $\sigma$ should remain lower or equal to 0.1, otherwise, we sample too far from $\mathbf{z}_0$ and generate completely new cells, breaking the correlation with the pre-masking gene expression values and, therefore, violating the purpose of imputation. Collectively, our results serve as additional evidence of the value of our model in community-oriented application settings.

---

> > > ### Author Response · Authors · 2024-11-24
> > >
> > > Dear Reviewer,
> > >
> > > Thank you once again for your valuable feedback on our work. We have carefully considered the suggestions raised during the review process and incorporated them into an improved version of our paper. Specifically, we have added new sections on model runtime and missing value imputation to address the highlighted points.
> > >
> > > We hope these revisions adequately resolve the remaining concerns. If the reviewer finds that our response has satisfactorily addressed the feedback, we would be grateful if they might consider increasing their score to reflect this. Of course, we remain available and willing to address any further concerns or questions the reviewer may have.
> > >
> > > Thank you for your time and thoughtful review.
> > >
> > > Best regards,
> > >
> > > The Authors

---

> > > > ### Author Response · Authors · 2024-11-28
> > > >
> > > > Dear Reviewer eqfJ,
> > > >
> > > > Thank you again for providing your valuable feedback and enabling the improvement of our paper with additional results concerning:
> > > > * The addition of training and sampling runtime in comparison with baseline methods.
> > > > * The extension of the scope of our model to the task of missing gene imputation in single-cell RNA-seq.
> > > >
> > > > We hope these revisions sufficiently address the remaining concerns. Of course, we remain fully available and willing to engage with any further questions or concerns the reviewer may have until the end of the discussion period.
> > > >
> > > > Thank you again for your time and kind consideration.
> > > >
> > > > The Authors

---

### Official Review · Reviewer_DU3L · 2024-11-05

**Soundness:** 3
**Presentation:** 3
**Contribution:** 3
**Rating:** 8
**Confidence:** 3

**Summary:**

The paper presents conditional flow-based generative models for single-cell RNA-seq and accessibility data. Single cell data is generally sparse, noisy, and has high feature variance. The authors suggest a flow matching based approach as a more expressive, and consistent generative model compared to VAEs, and GANs for generating synthetic cells. They also present a compositional variant of classifier-free guidance for flow-based models to allow conditioning on various attributes. Finally, they evaluate the model on two downstream tasks: (1) generating synthetic samples of rare cell-types and using them for data-augmentation,  (2) leveraging CFGen for batch correction.

**Strengths:**

1. The paper addresses an important problem in single-cell data generation by generating raw count values, and further extending this to multimodal generation.
2. The paper is well-written, and the authors convey major limitations of their model clearly.
3. The results show that CFGen is able to capture characteristics of the training dataset and generate single cell data with similar statistical properties.
4. They also show the effectiveness of generating rare cell-types to improve classification performance for other models.

Post Rebuttal comments:

The authors have addressed my concerns regarding the presentation. They have also added the additional details I addressed in the weaknesses below. After going through their responses to other reviewers, I believe the paper will be a valuable addition to ICLR. I am raising my score to accept.

**Weaknesses:**

1. Fig 3. is not really clear to me. Firstly, I suggest adding contrasting colors for points representing generated and real data. Secondly, what are the red points representing? I also suggest perhaps adding a quantitative metric (perhaps a oracle model that predicts the attributes) as well.
2. I also suggest removing the bars from Fig. 2b as they make it hard to observe the overlapping density curves which are easier to infer from.
3. For Sec 5.2, it might be worthwhile to also add a comparison with CFGen just trained on RNA-data in order to measure the effects of using multimodal data for training.
4. A comparison of inference times might also be useful in this case, especially to compare scDiffusion and CFGen, since both require multiple time steps. Adding approximate training times for each of the comparable models would also be valuable.
5. Fig.4 should also report the raw accuracy numbers for each of the cell-types to evaluate the effect of CFGen,

**Questions:**

See weaknesses above.

---

> ### Author Response · Authors · 2024-11-21
> **Rebuttal 1**
>
> We are thankful for the time DU3L invested in reviewing our paper. We are glad to hear the reviewer found our contribution solid and the paper well written. We appreciate the constructive criticism and updated our paper in the direction of the suggested changes.
>
> > Fig 3. is not really clear to me. Firstly, I suggest adding contrasting colors for points representing generated and real data. Secondly, what are the red points representing?
>
> We acknowledged the reviewer's point of view and modified the figure accordingly. We introduced a stronger red-blue contrast between real and generated cells and used the same color scheme consistently across the whole figure to represent real and generated cells. Now both in the zero-guidance case and in the guided generation scheme blue points represent real cells and red points generated cells. We also made the legend bigger.
>
> We briefly summarize the plot in case it remains unclear. On the left, enclosed by the boxes, we show the qualitative generative performance of the model when setting guidance parameters to 0. Such an approach allows a faithful modeling of the whole single cell distribution since the flow is not guided by any attribute. Next to the boxes, we show the effect of increasing a single guidance parameter over the counterpart, hence generating cells with increased specificity from an intersection of attributes. The middle plots with points highlighted in red are the generated cells for a certain guidance scheme (hence, specific guidance weights). On the opposite sides of such plots, with cells colored in blue, we highlight the real points coming from the two categories used for guidance. We include the latter plots as a reference since we expect guidance parameters to become increasingly good at modeling intersections of such attributes.
>
> > I also suggest perhaps adding a quantitative metric (perhaps an oracle model that predicts the attributes) as well.
>
> We added such a metric to Tab. 12 in the Appendix. We train as an oracle a 3-layer MLP classifier with a softmax head on the real data. Specifically, we derive one classifier per guidance attribute. Upon generation with different guiding schemes, we apply the two classifiers to the generated cells. We expect that, when the model is guided on a single attribute only (hence, the counterpart attribute has guidance strengths $\omega=0$), the oracle only assigns high probability to the class involved in guidance. This is indeed what occurs in Tab. 12, in the first row, only the cell type (NeurIPS) and tissue (C. Elegans) attributes are used for guidance. Therefore, the oracle assigns the cells generated under this scheme a high probability of being part of the guiding biological annotation class, while the probability for the mouse ID and donor classes we chose is low (the model did not use them to guide generation). Upon increasing the donor and mouse ID weights, the oracle predicts generated cells to be part of the guiding classes from both attributes with high probability.
>
> > I also suggest removing the bars from Fig. 2b as they make it hard to observe the overlapping density curves which are easier to infer from.
>
> We modified the plot according to the suggestion. The new version of Fig.2 can be found in the updated manuscript.
>
> > For Sec 5.2, it might be worthwhile to also add a comparison with CFGen just trained on RNA-data in order to measure the effects of using multimodal data for training.
>
> We added CFGen trained on RNA data only to Tab. 2. The model achieved top performance in terms of RBF-MMD and second-best performance (below its multimodal counterpart) in terms of Wasserstein-2 distance. Hence, multi-modal generation on PBMC10k performs on par with its RNA-only counterpart. From our results, we infer that modeling multi-modal data does not hamper the performance of our model despite the increased amount of information to synthesize.

---

> > ### Author Response · Authors · 2024-11-21
> > **Rebuttal 2**
> >
> > > A comparison of inference times might also be useful in this case, especially to compare scDiffusion and CFGen, since both require multiple time steps. Adding approximate training times for each of the comparable models would also be valuable.
> >
> > We added a new Appendix section (H.1) performing a detailed breakdown of CFGen's runtime in comparison to other models. In Fig. A1 we show what hyperparameters influence the generation runtime the most. Meanwhile, in Tab. 5 we illustrate the runtime per training epochs and in Tab. 6 the time (in seconds) required for sampling. Both tables were evaluated across all four datasets benchmarked in Section 5.1.
> >
> > From the sampling runtime in Tab. 6, one can infer that VAE-based models (scVI and MultiVI) are generally faster. However, we highlight that they are inherently less expressive and  worse performing than CFGen, especially when it comes to large datasets (see Tab. 1 and Fig. A7). Crucially, CFGen is **drastically** faster than scDiffusion, speeding upsampling by orders of magnitude. This happens for the following reasons:
> > * We use fewer simulation steps than scDiffusion (5-10 in CFGen, >1000 for scDiffusion) while gaining superior empirical and quantitative results.
> > * We use a much lower dimensional latent space (50-100 dimensions for CFGen, 1000 dimensions for scDiffusion as recommended in the manuscript).
> > * scDiffusion uses classifier-based guidance while we use classifier-free guidance. Hence, our performance is not influenced by the gradient of a classifier's prediction for each step.
> >
> > Strikingly, CFGen can reliably sample 1M cells in around 15 seconds (Fig. A1) and generate comprehensive atlases with >500,000 cells like the HLCA in 8 seconds.
> >
> > For fairness, we highlight that the speedup still depends on the batch size one can fit into memory during sampling (10k cells in our case).
> >
> > > Fig.4 should also report the raw accuracy numbers for each of the cell types to evaluate the effect of CFGen.
> >
> > We added two tables to the Appendix reporting raw classification performance before and after augmentation together with the cell type frequency for both the PBMC covid (Tab. 9) and HLCA (Tab. 10) datasets. In both tables, we highlight the highest value between before and after augmentation accuracy for each row. Notably, in most cases, augmentation induces an improvement in per-cell-type accuracy. Some classification performances increase significantly upon augmentation. We highlight the following examples:
> > * Dendritic cells 0.47 -> 0.80 (PBMC)
> > * Innate lymphoid cells 0.21 -> 0.40 (PBMC)
> > * Tracheobronchial goblet celL 0.00 -> 0.50 (HLCA)
> > * Stromal cell 0.30 -> 0.60 (HLCA)
> > * Brush cell of tracheobronchial tree 0.35 -> 0.88 (HLCA)
> > * Dendritic cell 0.46 -> 0.70 (HLCA)
> >
> > For the remaining scores, we kindly refer the reviewer to Appendix H.6.
> >
> > We thank once again Reviewer DU3L for their consideration and remain available to provide additional clarifications.

---

> > > ### Author Response · Authors · 2024-11-24
> > >
> > > Dear Reviewer,
> > >
> > > We are grateful for the time you have invested in reviewing our work. Your suggested improvements made our presentation more solid and comprehensive, strengthening the quality of our paper.
> > >
> > > Since we are approaching the end of the discussion period, we would like to make sure our rebuttal exhaustively covered all the remaining concerns raised in the review about figure improvement and integrations to the result section.
> > >
> > > If our response has successfully addressed the insightful points raised in the review, we kindly ask whether the reviewer could consider increasing their score. Of course, we are happy to address any additional questions or concerns as needed.
> > >
> > > Thank you once again for the thorough feedback and for your time.
> > >
> > > Best regards,
> > >
> > > The Authors

---

> > > > ### Author Response · Authors · 2024-11-28
> > > >
> > > > Dear Reviewer DU3L,
> > > >
> > > > We thank you again for investing the time to review our paper and provide thorough feedback. Following your insightful suggestions, we revised our manuscript to include:
> > > > * A better version of figures 2 and 3.
> > > > * An evaluation of the multi-attribute approach done through an oracle model.
> > > > * The comparison of multimodal CFGen with its RNA-only version on PBMC10K.
> > > > * A detailed breakdown of sampling and training runtimes in comparison with baselines.
> > > > * Raw accuracy values of the classifier model.
> > > >
> > > > As we approach the conclusion of the discussion period, we would greatly appreciate hearing whether the reviewer feels their concerns have been fully addressed. We remain fully available to provide clarifications or answer any additional questions they may have.
> > > >
> > > > Again, thank you very much for your valuable time.
> > > >
> > > > Best regards,
> > > >
> > > > The Authors

---

> > > > > ### Comment · Reviewer_DU3L · 2024-12-02
> > > > > **Thank you for the response**
> > > > >
> > > > > I thank the authors for their response. Overall, CFGen appears to be competitive with existing baselines like scVI, and scDiff. I appreciate their efforts with adding the new results, and fixing the presentation of the paper. I am keeping my score for now, but am willing to increase it after discussing with the other reviewers.

---

> > > > > > ### Author Response · Authors · 2024-12-02
> > > > > >
> > > > > > Dear Reviewer DU3L,
> > > > > >
> > > > > > Thank you once again for reviewing our paper and providing thoughtful feedback on our rebuttal. We are pleased to hear that the reviewer recognizes the improvements in the paper's presentation and the enhanced quality of our new results.
> > > > > >
> > > > > > In light of the reviewer's comments, we would like to kindly emphasize that CFGen is more than just competitive when compared to the cited baselines. We summarize the reasons supporting our claim below:
> > > > > >
> > > > > > - Aside from **consistently outperforming** competing models on most datasets and metrics for both single-modality (Table 1) and multi-modality (Table 2) generation, CFGen is the **only model** that produces reliable results on large datasets, such as Tabula Muris (>200k cells) and HLCA (>500k cells), as shown in Fig. A7.
> > > > > > - Methodologically, CFGen is the **only model enabling multi-attribute generation** via the guidance mechanism. This approach demonstrated an improvement in batch correction performance not only on scVI but also on its more effective variants. To our knowledge, multi-attribute classifier-free guidance in Flow Matching has not been explored before in machine learning.
> > > > > > - CFGen is also the **only model among the three** providing reliable augmentations that enhance cell type classification performance (Fig. A11).
> > > > > > - From a methodological perspective, CFGen introduces a superior model to scVI for generating negative binomial counts from noise. Its unique factorization **overcomes the limitations of disjoint sampling** between the size factor and the latent variable (see the second answer to Reviewer FqvM).
> > > > > > - Compared to scDiffusion, we have shown that CFGen offers better guarantees for reproducing realistic properties of single-cell data (Sec. 5.1) without compromising the expressiveness of ODE-based generative models.
> > > > > >
> > > > > > In summary, we believe that CFGen represents a significant improvement over existing baselines by enabling novel applications of generative models in single-cell RNA-seq through methodological novelty, while also boosting performance on existing tasks.
> > > > > >
> > > > > > We sincerely thank Reviewer DU3L for their thoughtful and constructive feedback which has greatly improved this work, and we hope that our additional clarifications will support a more favorable assessment of our manuscript.
> > > > > >
> > > > > > The Authors

---

### Official Review · Reviewer_kEhi · 2024-11-07

**Soundness:** 2
**Presentation:** 3
**Contribution:** 3
**Rating:** 8
**Confidence:** 4

**Summary:**

This paper proposes CFGen, which is a latent flow-matching generative model for single-cell data, where the latent space is first learned by an autoencoder. To capture statistical properties specific to single-cell data, the autoencoders learn to decode the parameters of a negative binomial distribution and Bernoulli distribution, for RNA-seq and ATAC-seq data, respectively. Conditional generation is achieved through classifier guidance. Empirical results demonstrate that CFGen outperform other single-cell generative models in terms of (1) data generation to approximate the real data distribution, (2) data generation for rare cell type classification, and (3) batch correction.

**Strengths:**

- Adapting flow matching for single-cell data generation is a novel contribution.
- The proposed framework CFGen can be easily adapted for different uni- and multi-modal scenarios, as long as there are modality-specific autoencoders with a common latent space.

**Weaknesses:**

- scVI should be included as a baseline in Figure 2 because scVI accounts for overdispersion and zero inflation, whereas the current baselines in Figure 2 (scDiffusion and scGAN) do not.
- For downstream applications that rely on conditional generation, it is unclear how the classifier guidance strength is determined.
- Quantitative results are lacking when evaluating the compositional classifier guidance in Section 5.3. The change in MMD and WD with respect to the target distribution when increasing guidance strength can suffice.

**Questions:**

- For batch correction, is CFGen's performance (in terms of the Batch and Bio scores) sensitive to varying the guidance parameters? How does one tune the guidance parameters in practice?
- For cell type classification, simple models such as logistic regression (with or without regularization) are often used. Does data augmentation with CFGen improve performance for a logistic regression model?

---

> ### Author Response · Authors · 2024-11-21
> **Comment 1**
>
> We want to thank the reviewer for their feedback and suggestions, which will surely contribute to improving the quality of our experimental validation. We also thank the reviewer for their positive comments about our contribution.
>
> > scVI should be included as a baseline in Figure 2.
>
> We include the comparison with scVI in Appendix H.2 of the revised manuscript. More specifically, we provide two new pieces of information:
> * The plot comparing the distribution of the number of zeros per cell and mean-variance trend between real and generated cells by CFGen and scVI (Fig. A2).
> * Quantitative metrics to evaluate how well different models approximate the sparsity and overdispersion of the real data (Tab. 7).
>
> As you can see from Fig. A2, both scVI and CFGen model the sparsity and overdispersion characteristics solidly in the considered datasets. This is expected since both use a decoding mechanism optimized under a negative binomial likelihood model. However, note that capturing the sparsity and overdispersion trend does not necessarily boil down to modeling the whole transcription state properly. Indeed, in Fig. A7 (Appendix) we show that scVI struggles to retrieve a realistic single-cell representation on larger datasets, due to the lower flexibility of its generative model (see Tab. 1 for quantitative metrics confirming this aspect). On the contrary, CFGen closely samples from the data distribution.
>
> We complement Fig.2 by evaluating the following quantitative metrics for each generative model:
> * The 1D Wasserstein-2 distance between the vectors of per-cell number of zeroes from real and generated data.
> * The 1D Wasserstein-2 distance between the empirical mean-variance ratio vectors of real and generated cells.
>
> Note that we have to use a distributional distance because generated cells are not the same "items" as real cells but new objects, therefore we cannot evaluate a correlation between them. The lower such metrics get, the more closely a model's sparsity and mean-variance ratio resemble the one from the data. Results in Tab. 7 report that CFGen is the best model at approximating sparsity and overdispersion on three datasets out of four.
>
> > It is unclear how the classifier guidance strength is determined.
>
> The only downstream task where guidance is used is batch correction on the NeurIPS and C.Elegans datasets in Section 5.5. In Appendix H.9 we provide an intuition of our selection process. In batch correction, cells are transported to noise and then back again to data guided by a biological and a target batch covariate. The stronger the guidance strength parameters $\omega_{\mathrm{bio}}$ and $\omega_{\mathrm{batch}}$ the more biological conservation and batch conversion will be emphasized.
>
> If one merely observes the scIB metric in  Tab. 13 computed over different guidance strength parameters, they will choose the highest possible guidance strengths, since they provide the best aggregation within cell types and batches. However, in Fig. A16a-b and A17a-b, we show that scIB metrics could be misleading and should be accompanied by qualitative evaluation. Indeed, increasing guidance parameters too much in the translation task leads to an unnatural collapse of the variability in the data beyond the one explained by batch and biological annotations. On another note, we found that guidance strength parameters surrounding values of 1 and 2 are sufficient at both preserving signal in the data and performing correction without over-squashing the cell representations. An example of unwanted effects is presented in Fig. A16, where biological preservation results in unnatural clustering for both datasets.
>
> Finally, it is important to consider the extent of the batch effect present in the data. For example, in C. Elegans the batch effect is mild, therefore we select the parameters $\omega_{\mathrm{bio}}=2,\omega_{\mathrm{batch}}=1$ since they provide better performance than  $\omega_{\mathrm{bio}}=1, \omega_{\mathrm{batch}}=2$ and $\omega_{\mathrm{bio}}=1, \omega_{\mathrm{batch}}=1$ (Tab. 13). In the NeurIPS dataset, we observe the opposite effect and therefore select $\omega_{\mathrm{bio}}=2, \omega_{\mathrm{batch}}=1$, since as soon as $\omega_{\mathrm{bio}}>1$ we obtain the unnatural biological structure in Fig A17c, which violates smooth temporal single-cell trajectories.
>
> In conclusion, we recommend first evaluating the extent of batch effect in the data and then sweeping over combinations of guidance weights, selecting the configuration that achieves the best scIB metric values without inducing unrealistic single-cell representations.

---

> ### Author Response · Authors · 2024-11-21
> **Comment 2**
>
> > For batch correction, is CFGen's performance (in terms of the Batch and Bio scores) sensitive to varying the guidance parameters? How does one tune the guidance parameters in practice?
>
> Please, see the answer above.
>
> > Quantitative results are lacking when evaluating the compositional classifier guidance in Section 5.3. The change in MMD and WD with respect to the target distribution when increasing guidance strength can suffice.
>
> We include the suggested results in Fig. A15 (Appendix H.8). In the experiment, we increase the guidance parameters for both considered attributes in parallel from 0.1 to 2.5 and evaluate how well the generated cells approximate the real cells at the intersection of the two labels. Notably, increasing the guidance weights improves modeling the combinations of attributes, with both lower MMD and Wasserstein-2 distance values.
>
> > Does data augmentation with CFGen improve performance for a logistic regression model?
>
> We explored such a direction in the new Fig. A12. More in detail, we apply the setting described for scGPT in Section 5.4 to CellTypist [1], a famous cell type annotation model based on logistic regression. Interestingly, a similar trend is observed for the linear classifier as the one reported for scGPT, where the rare cell type classification accuracy on unseen patients increases upon data augmentation. This is particularly evident in the PBMC covid dataset where the majority of cell types exhibit a boost in predictive performance over held-out patients. In HLCA the overall trend still favors an improvement in rare cell type classification, though less pronounced and consistent than the PBMC covid dataset. Overall, our results suggest our augmentation strategy can apply to multiple classifiers.
>
> We would like to thank again Reviewer kEhi for their consideration and remain available for further clarifications.
>
> [1] Cippà, Pietro E., and Thomas F. Mueller. "A First Step Toward a Cross-tissue Atlas of Immune Cells in Humans." Transplantation 107.1 (2023): 8-9.

---

> > ### Author Response · Authors · 2024-11-24
> >
> > Dear Reviewer,
> >
> > Thank you for the time and effort you have dedicated to reviewing our work. Your thoughtful feedback has been instrumental in enhancing the clarity and rigor of our experimental evaluation.
> >
> > As we near the end of the discussion period, we wanted to ensure that our previous responses adequately addressed the reviewer's concerns with additional experimental evidence and details of our approach to model selection. We hope our clarifications have resolved any remaining uncertainties about our contribution.
> >
> > We would be grateful if the reviewer could kindly consider increasing their score if our response has succeeded in addressing all the great points raised in the review. Of course, we remain happy to address any additional concerns if needed.
> >
> > Thank you once again for your valuable input.
> >
> > Best regards,
> >
> > The Authors

---

> ### Author Response · Authors · 2024-11-28
>
> Dear Reviewer kEhi,
>
> Thank you once again for taking the time to provide thoughtful feedback on our work. Your insightful comments have significantly contributed to enhancing our paper in the following ways:
> * Providing a more comprehensive set of baselines.
> * Clarifying model selection practices in the batch correction task.
> * Demonstrating promising generative performance results in the multi-attribute generation task.
> * Offering additional evidence that augmentation with CFGen improves rare cell type classification using a logistic regression model.
>
> We hope these revisions address the key points the reviewer raised and would greatly appreciate hearing whether the reviewer believes their concerns have been fully addressed. As always, we are happy to discuss any further remaining questions or suggestions.
>
> Thank you again for reviewing our work.
>
> Best regards,
>
> The Authors

---

> > ### Comment · Reviewer_kEhi · 2024-12-03
> >
> > The additional experiments and results have addressed my concerns. I especially appreciate the authors for describing the process for selecting the classifier guidance strength. I encourage the authors to include the following description in their revised manuscript:
> > ```
> > In conclusion, we recommend first evaluating the extent of batch effect in the data and then sweeping over combinations of guidance weights, selecting the configuration that achieves the best scIB metric values without inducing unrealistic single-cell representations.
> > ```
> >
> > Overall, although the ideas of latent-space flow matching and classifier guidance are not necessarily novel from a machine learning perspective, their adaption to single-cell biology is novel for computational biology. Therefore, I increased my score from a 6 to 8.

---

> ### Author Response · Authors · 2024-12-03
>
> Dear Reviewer kEhi,
>
> Thank you for your thoughtful feedback and acknowledging our rebuttal and contribution to the field. We sincerely appreciate your decision to increase your score to acceptance and your recognition of our work's value to computational biology.
>
> We are also grateful for your suggestion to update the conclusion to reflect our new batch correction results. We will incorporate the provided text into the revised version of our manuscript.
>
> Thank you again for your support and contribution to improving our paper.
>
> Best regards,
>
> The Authors

---

### Author Response · Authors · 2024-11-21
**General comment**

We sincerely thank all the reviewers for their constructive feedback. As authors, we greatly appreciate the opportunity to enhance the scientific quality of our work by addressing reported inconsistencies, additional experiment requests, and general criticism. All experiments and specific suggestions were carefully addressed and will be discussed separately for each reviewer in the dedicated rebuttal sections.

Aside from providing key clarifications, our rebuttal output incorporates a considerable number of new experiments including:
* A detailed runtime analysis in comparison with baseline models (Appendix H.1).
* Quantitative evaluations of the multi-attribute setting (Appendix H.8).
* Cell type classification performance improvement using linear models (Appendix H.6).
* Comparison with CFGen trained on scRNA-seq only in the multi-modal setting (Tab. 2).
* The application of CFGen to the missing value imputation task (Appendix H.7).
* A deeper insight into the guidance strength selection for batch correction (Appendix H.9).

We additionally supplied missing details and comparisons (e.g., the raw cell type classification accuracy before and after augmentation by CFGen) and improved the figures where suggested.

All edits applied to the main manuscript are highlighted in blue.

We thank again both the reviewers and the area chair for their valuable consideration and look forward to follow-up discussions.

Best regards,

The Authors

---

### Author Response · Authors · 2024-12-03
**Summary of Discussion Outcomes and Manuscript Improvements**

Dear AC and Reviewers,

As the interaction period with reviewers nears its conclusion, we wish to provide a concise summary of the discussion outcomes and the improvements made to our manuscript in response to the insightful feedback received:

- **Reviewer kEhi:**
  We addressed additional concerns by extending model comparisons with scVI, elaborating on parameter selection for batch correction tasks, and demonstrating how multi-attribute guidance improves cell generation at the intersection of attributes. Evidence of CFGen's benefits for data augmentation in a linear classifier was also included. Following our rebuttal, Reviewer kEhi acknowledged the changes by increasing their score and suggesting further text additions, which we will incorporate.

- **Reviewer DU3L:**
  We clarified Figures 2b and 3, demonstrated that the multimodal component does not impact scRNA-seq generation quality (Tab. 2), and provided evidence of expected performance for multi-attribute guidance via classification of intermediate outputs at varying guidance strengths. Additional details on runtime and raw accuracy values for cell type classification pre- and post-augmentation were also provided. Reviewer DU3L acknowledged our responses and confirmed that their remaining concerns were addressed.

- **Reviewer eqfJ:**
  We included a detailed breakdown of model runtime, analyzing hyperparameter effects and benchmarking against baselines, which showed significant speedup over scDiffusion, our main ODE-based competitor. Additionally, we extended the scope to gene imputation, reporting promising results. While Reviewer eqfJ did not provide further responses, **all experimental concerns** raised during the review phase were thoroughly addressed.

- **Reviewer FqvM:**
  We elaborated on our contributions, including the choice of factorization, the conditional independence assumption, and additional details on the guidance approach. We also provided reasoning for scDiffusion's underperformance in our setting. Reviewer FqvM acknowledged our response and raised their score above the acceptance threshold.

We are grateful to the AC for their continued support throughout the review and discussion process. We also extend our sincere thanks to the reviewers for their valuable time and thoughtful suggestions, which significantly improved our manuscript.

Best regards,

The Authors

---

### Meta-Review · Area_Chair_2HZT · 2024-12-31

**Metareview:**

The paper presents a flow-based conditional generative model for single-cell RNA sequence data. The reviewers were generally positive about the strong: the adaptation of flow-matching to this setting is novel, the framework seems easily adaptable to many realistic downstream scenarios, and the empirical results are solid. While the paper's original presentation had a few weaknesses, the authors have adequately addressed these during the rebuttal period. Given this, I am enthusiastically recommending acceptance. Please incorporate the reviewers' feedback carefully in the final version.

**Additional Comments On Reviewer Discussion:**

The authors posted numerous comments during the rebuttal period. The comments are highly appreciated; while they didn't trigger substantial discussion, they caused multiple reviewers to increase their scores.

---

### Decision · Program_Chairs · 2025-01-22

Accept (Poster)